

# Bethe Ansatz inside Calogero-Sutherland models

Gwenaël Ferrando[1°], Jules Lamers[2,3*§], Fedor Levkovich-Maslyuk[2¶] and Didina Serban[2]

**1** School of Physics and Astronomy, Tel Aviv University, Ramat Aviv 69978, Israel
**2** Université Paris–Saclay, CNRS, CEA, Institut de Physique Théorique,
91191 Gif-sur-Yvette, France
**3** Deutsches Elektronen-Synchrotron DESY, Notkestraße 85, 22607 Hamburg, Germany

⋆ jules.lamers@glasgow.ac.uk

## Abstract

We study the trigonometric quantum spin-Calogero–Sutherland model, and the Haldane–Shastry spin chain as a special case, using a Bethe-Ansatz analysis. We harness the model's Yangian symmetry to import the standard tools of integrability for Heisenberg spin chains into the world of integrable long-range models with spins. From the transfer matrix with a diagonal twist we construct Heisenberg-style symmetries (Bethe algebra) that refine the usual hierarchy of commuting Hamiltonians (quantum determinant) of the spin-Calogero–Sutherland model. We compute the first few of these new conserved charges explicitly, and diagonalise them by Bethe Ansatz inside each irreducible Yangian representation. This yields a new eigenbasis for the spin-Calogero–Sutherland model that generalises the Yangian Gelfand–Tsetlin basis of Takemura–Uglov. The Bethe-Ansatz analysis involves non-generic values of the inhomogeneities. Our review of the inhomogeneous Heisenberg xxx chain, with special attention to how the Bethe Ansatz works in the presence of fusion, may be of independent interest.

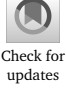

## Contents

° Present address: Bethe Center for Theoretical Physics, Universität Bonn, Wegelerstr. 10, D-53115, Germany.
§ Present address: School of Mathematics and Statistics, University of Glasgow, University Place, Glasgow G12 8QQ, UK.
¶ Present address: Centre for Mathematical Science, City St George's, University of London, Northampton Square, EC1V 0HB, London, UK.

## 1 Introduction

Long-range interacting spin systems appear naturally in a broad range of physical contexts, from experiments with cold atoms to high-energy theory [1, 2]. Yet on the theoretical side they have received much less attention than their nearest-neighbour counterparts.

In this paper we will consider three integrable long-range models which, as we will see, are all interrelated. The first one has actually been studied for nearly half a century, although it is usually not explicitly thought of as a long-range model: the inhomogeneous Heisenberg spin chain. It naturally arises from the viewpoint of the six-vertex model (cf. Baxter's $Z$-invariant model [3]) and makes an appearance in the Bethe/gauge correspondence [4], where it corresponds to a certain $\mathcal{N} = 2$ supersymmetric gauge theory (with 'twisted masses' that relate to the inhomogeneities). The inhomogeneous Heisenberg chain is an important example of a quantum-integrable system, with an underlying Yangian structure that provides its commuting charges (from the transfer matrix, or Bethe algebra) as well as a way of diagonalising them (by algebraic Bethe Ansatz). Yet, except for some special cases, one usually does not

think of it as a *bona fide* spin chain, and the inhomogeneity parameters are rather seen as a technical tool. For instance, they are crucial for the Izergin–Korepin approach to computing the domain-wall partition functions, Gaudin norms, and Slavnov scalar products of Bethe vectors [5,6]. Other applications of inhomogeneities are the proofs of completeness of the Bethe Ansatz [7–9] and of some of the Razumov–Stroganov conjectures [10]. In addition, in special semiclassical limits, inhomogenous spin chains give rise to the Gaudin Hamiltonians [11]. For most other physical applications one eventually takes the homogeneous limit to restore periodicity. One can also consider special repeating values of the inhomogeneities, e.g. staggered (alternating) values [12–15] or other periodic values [16,17], which give access to a broader range of conformal field theories in a suitable scaling limit.

The second long-range model will be the main object of our study: the quantum trigonometric Calogero–Sutherland model with particles that have spins. For a system with $N$ particles and 'reduced' coupling constant $\beta$, the Hamiltonian is [18–20]

$$\widetilde{H} = -\frac{1}{2} \sum_{i=1}^{N} \partial_{x_i}^2 + \beta \sum_{i<j} \frac{\beta \mp P_{ij}}{4 \sin^2[(x_i - x_j)/2]}, \tag{1}$$

where the upper (lower) sign corresponds to bosonic (respectively fermionic) statistics, and $P_{ij}$ is the spin permutation operator. This quantum many-body system is also integrable, has eigenvectors containing Jack polynomials with prescribed (anti)symmetry, and a deep representation-theoretic structure [21, 22] which in particular provides a representation of the Yangian that is very different from the usual one for Heisenberg spin chains. The quantum determinant, i.e. the centre of the Yangian, generates the commuting Hamiltonians of the spin-Calogero–Sutherland model, which means that the spin symmetry is enhanced to Yangian *symmetry*. (In contrast, for the Heisenberg spin chain the Yangian can be used to move between eigenvectors with different energies, as in the algebraic Bethe Ansatz.) The Yangian structure was studied in detail by Takemura and Uglov [23–25]. A closely related property is that the spin-Calogero–Sutherland model is superintegrable, see [26], which means that it has (even) more commuting charges than a typical integrable system such as the Heisenberg spin chain or scalar Calogero–Sutherland model. At the level of the spectrum, the superintegrability shows up as extra degeneracies compared to the spinless case. These degeneracies are controlled by the Yangian, which combines the degenerate states into an irreducible representation.

One of our main motivations for studying the spin-Calogero–Sutherland model is its connections to the third long-range model: the Haldane–Shastry spin chain, with Hamiltonian [27,28]

$$H^{\text{HS}} = \sum_{i<j} \frac{1 + P_{ij}}{4 \sin^2\left[\frac{\pi}{N}(i-j)\right]}. \tag{2}$$

It exhibits fractional (exclusion) statistics [29–31] and can be viewed as the $SU(2)_{k=1}$ Wess–Zumino–Witten model on the lattice [32–35]. The Haldane–Shastry chain arises from (1) in a special 'freezing' limit [36–38] and inherits various special properties along the way. Amongst others, $H^{\text{HS}}$ has very simple eigenvalues and very high degeneracies [30], which are to a large extent due to its Yangian symmetry [32,38]. Its (Yangian highest-weight) wave functions are given by certain symmetric Jack polynomials, which are indirectly derived from freezing [38]. Higher Hamiltonians follow from freezing too [39,40]. Although the freezing procedure often starts from the bosonic spin-Calogero–Sutherland model [36–38], two of us showed [41] that the *fermionic* case naturally accounts for the form of the Haldane–Shastry wave functions with Yangian highest weight, and used it to prove a claim from [38] about the spin-chain eigenvectors for higher rank. We refer to the introduction of [41] for a more in-depth survey of different connections between the spin-Calogero–Sutherland model and Haldane–Shastry spin chain.

The following higher Hamiltonians of the Haldane–Shastry spin chain were known. Inozemtsev [42] identified[1]

$$H_3^{\text{HS}} = \sum_{i,j,k=1}^{N}{}' \frac{P_{ij}P_{jk}}{\sin\left[\frac{\pi}{N}(i-j)\right]\sin\left[\frac{\pi}{N}(j-k)\right]\sin\left[\frac{\pi}{N}(k-i)\right]}, \tag{3}$$

$$I_3^{\text{HS}} = \sum_{i,j,k=1}^{N}{}' \cot\left[\tfrac{\pi}{N}(i-j)\right]P_{ij}P_{jk}, \tag{4}$$

where the prime indicates that equal values of the indices are omitted from the sum. Next,

$$H_4^{\text{HS}} = \sum_{i,j,k,l=1}^{N}{}' \frac{P_{ij}P_{jk}P_{kl}}{\sin\left[\frac{\pi}{N}(i-j)\right]\sin\left[\frac{\pi}{N}(j-k)\right]\sin\left[\frac{\pi}{N}(k-l)\right]\sin\left[\frac{\pi}{N}(l-i)\right]} - 2\sum_{i,j=1}^{N}{}' \frac{P_{ij}}{\sin^4\left[\frac{\pi}{N}(i-j)\right]} \tag{5}$$

was empirically found in [32] and properly derived in [40]. Like $H_2^{\text{HS}} = H^{\text{HS}}$, the operators $H_3^{\text{HS}}$ and $H_4^{\text{HS}}$ belong to the family of Calogero–Sutherland-style charges, which commute with each other and the Yangian. Further charges of this type can be constructed following [40]. Inozemtsev's $I_3^{\text{HS}}$ is somewhat of an outlier: it commutes with all of these $H_n^{\text{HS}}$ and $\mathfrak{su}_2$, but *not* with (the rest of) the Yangian.[2]

**In this paper** we import the toolkit of Heisenberg integrability into the world of the spin-Calogero–Sutherland model and Haldane–Shastry chain. We exploit the Yangian symmetry to construct additional commuting charges of the spin-Calogero–Sutherland model. Namely, we will use a transfer matrix to construct symmetries that form (a representation of) a maximal abelian algebra of the Yangian, called the Bethe algebra. The first charge that we obtain in this way, besides the (total) momentum operator $-i\sum_j \partial_{x_j}$ and the Hamiltonian (1), is

$$\frac{\kappa + \kappa^{-1}}{2}\sum_{i<j}P_{ij} + \frac{\kappa - \kappa^{-1}}{2}\left(\frac{1}{\beta}\sum_{i=1}^{N}\sigma_i^z z_i\,\partial_{z_i} + \sum_{i<j}\left(\frac{z_i\,\sigma_j^z - z_j\,\sigma_i^z}{z_i - z_j}P_{ij} - \frac{1}{2}\frac{z_i + z_j}{z_i - z_j}(\sigma_i^z - \sigma_j^z)\right)\right), \tag{6}$$

where $\kappa$ is a twist parameter. Away from the periodic case $\kappa = \pm 1$, where (6) is just the quadratic Casimir, our extra *Heisenberg-style charges* commute with the $H_n^{\text{HS}}$, but not with the Yangian, showing that they go beyond the usual spin-Calogero–Sutherland hierarchy. Via the freezing procedure we in particular obtain extra charges of the Haldane–Shastry chain. An example of such a charge is Inozemtsev's (4). We thus provide a systematic way to construct a whole hierarchy of Heisenberg-style charges beyond (4), which moreover admits a deformation by the twist parameter $\kappa$. For example, the frozen version of (6) at $\kappa = i$ reduces to

$$\sum_{i<j}^{N} \frac{e^{i\pi(i-j)/N}\sigma_j^z - e^{i\pi(j-i)/N}\sigma_i^z}{\sin\left[\frac{\pi}{N}(i-j)\right]}P_{ij}, \tag{7}$$

which does not commute with the usual spin raising and lowering operators.

Our refinement of the conserved charges enables us to simultaneously diagonalise all of them by the algebraic Bethe Ansatz and construct a new eigenbasis for the spin-Calogero–Sutherland model. By including a (diagonal) twist, our construction generalises the Yangian Gelfand–Tsetlin eigenbasis constructed by Takemura and Uglov [23]. In more detail, the

---

[1]By antisymmetry of the coefficients, $P_{ij}P_{jk}$ can be replaced by $[P_{ij},P_{jk}]/2 = \vec{\sigma}_i\cdot(\vec{\sigma}_j\times\vec{\sigma}_k)/4i$. In particular, $I_3^{\text{HS}} = -i\vec{S}\cdot\vec{\Lambda}$, where $\vec{S} = \sum_{j=1}^{N}\vec{\sigma}_j/2$ are the $\mathfrak{su}_2$ operators, and $\vec{\Lambda} = \sum_{i<j}^{N}\cot\left[\frac{\pi}{N}(i-j)\right]\vec{\sigma}_i\times\vec{\sigma}_j$ is Haldane's 'rapidity operator', cf. [39], whose components are, in turn, 4× the 'level-1' generators of the Yangian in Drinfeld's first realisation, cf. pp. 9–10 of [43] and §C.2 of [41].

[2]Fowler and Minnahan [39] constructed a family of operators including (4). While their next operator commutes with the $H_n^{\text{HS}}$, it does not seem to commute with (4) for $N \geq 7$.

eigenspaces of the spin-Calogero–Sutherland model, labelled by partitions with bounded multiplicities, are irreducible representations of the Yangian that were studied in detail in [23]. We reinterpret each such eigenspace as an 'effective' inhomogenous Heisenberg spin chain, with special values of the inhomogeneities. The Yangian highest-weight vector, which can be described explicitly in the coordinate basis [41], serves as the pseudovacuum of the effective spin chain. We then use the algebraic Bethe Ansatz to generate the full eigenspace of the Calogero–Sutherland model, and by freezing, of the Haldane–Shastry spin chain. Remarkably, the values of the inhomogeneities force us to consider cases where fusion occurs. Thus, we will first review how fusion works and point out various subtleties for the algebraic Bethe Ansatz in this situation. An interesting feature of our construction is that we apply the algebraic Bethe Ansatz starting from pseudovacua that generally do not have the simple form $|\uparrow \cdots \uparrow\rangle$.

Explicitly, the Bethe equations for the spin-Calogero–Sutherland model are as follows. The eigenspaces are labelled by partitions $\boldsymbol{\lambda} = (\lambda_1 \geqslant \cdots \geqslant \lambda_N) \in \mathbb{Z}^N$ with multiplicities $\leqslant 2$. The highest-weight vector occurs at $S^z = N/2 - M_{\boldsymbol{\lambda}}$, where $M_{\boldsymbol{\lambda}}$ is the number of repeats in $\boldsymbol{\lambda}$. Using this vector as pseudovacuum, the Bethe equations giving eigenvectors of the Heisenberg-style symmetries at $S^z = N/2 - M_{\boldsymbol{\lambda}} - M$ read

$$\kappa^2 \prod_{j \in I_{\boldsymbol{\lambda}}} \frac{u_m - \theta_j(\boldsymbol{\lambda}) + \frac{\mathrm{i}}{2}}{u_m - \theta_j(\boldsymbol{\lambda}) - \frac{\mathrm{i}}{2}} = \prod_{n(\neq m)}^{M} \frac{u_n - u_m + \mathrm{i}}{u_n - u_m - \mathrm{i}}, \qquad 1 \leqslant m \leqslant M \,, \tag{8}$$

where the product on the left-hand side runs over the $N - 2M_{\boldsymbol{\lambda}}$ parts of $\boldsymbol{\lambda}$ with multiplicity one. These are just the Bethe equations of the 'effective' Heisenberg spin chain, which has length $L_{\boldsymbol{\lambda}} = N - 2M_{\boldsymbol{\lambda}}$ and (imaginary) inhomogeneities $\theta_j(\boldsymbol{\lambda}) = -\mathrm{i}(\lambda_j/\beta + (N+1)/2 - j)$. In the freezing limit, $\beta \to \infty$, so $\theta_j(\boldsymbol{\lambda}) \to -\mathrm{i}(N + 1 - 2j)/2$ as long as the eigenspace remains nontrivial. The only information about $\boldsymbol{\lambda}$ that survives is the location of its repeated parts, which yields a so-called 'motif' of the Haldane–Shastry spin chain.

Our new Heisenberg-style symmetries provide a setting for developing separation of variables (SoV) for long-range spin chains that are richer than inhomogeneous Heisenberg spin chains. In addition, while we focus on the simplest case of $\mathfrak{sl}_2$ spins, our approach will extend to higher rank as well.

**Outline.** This paper is organised as follows. In Section 2 we review the algebraic Bethe Ansatz framework for Heisenberg XXX spin chains. We discuss the fusion procedure in detail, and pay special attention to nontrivial aspects of the Bethe Ansatz in this case. In Section 3 we recall the basics of the spin-Calogero–Sutherland model, its Yangian symmetry and its eigenspaces, reinterpreted as effective Heisenberg spin chains.

Section 4 contains our main results: the construction of a refined family of conserved charges for the spin-Calogero–Sutherland model and their diagonalisation by algebraic Bethe Ansatz. We analyse our construction in limiting cases, in particular including the Haldane–Shastry spin chain obtained by freezing, and illustrate it in an explicit example. Section 5 contains our conclusions.

Appendices A–D contain technical details and examples related to fusion for small systems.

## 2 How algebraic Bethe Ansatz works for inhomogeneous models

In this section we review the Bethe-Ansatz solution for the inhomogeneous XXX spin chain with a spin-1/2 representation at each site. We pay special attention to the subtleties of fusion, which are relevant for the long-range models that we focus on in the rest of the paper.

|  | center |  | max. ab. |  |
|---|---|---|---|---|
| **Quantum determinant** | $\subset$ | **Bethe algebra** | $\subset$ | **Yangian** |

- usual Calogero–Suth. charges:
  $\widetilde{P} = -\mathrm{i}\sum_j \partial_{x_j}$, $\widetilde{H}$ from (1), …
- Yangian *symmetry*:
  highly degenerate eigenspaces;
- *any* basis of Yangian irrep
  gives an eigenbasis

- Heisenberg-style charges:
  transfer matrix (with twist $\kappa$)
- no Yangian symmetry:
  degeneracies lifted
- distinguished eigenbasis:
  algebraic Bethe Ansatz

- monodromy matrix
  (nonstandard, but Hermitean)
- irreps: 'effective' Heis. chains
  with specific inhomogeneities
- explicit highest-weight vectors
  with partially symmetric Jacks

$$\downarrow \kappa \to \infty$$

**Gelfand–Tsetlin subalgebra**
- *A*-operator, quantum determinant
- simple model for Yangian structure
- distinguished eigenbasis:
  Gelfand–Tsetlin basis

Figure 1: Overview of the role of the Yangian, and distinguished subalgebras, for the spin-Calogero–Sutherland model. Traditionally one considers the Calogero–Sutherland-style charges (left) and the Yangian (right). In this work we focus on the Heisenberg-style charges (middle). The Haldane–Shastry spin chain inherits this setup upon freezing.

## 2.1 Heisenberg XXX spin chain

The Hamiltonian of the Heisenberg XXX spin chain for $L$ spin-1/2 sites is

$$H^{\mathrm{H}} = \sum_{i=1}^{L} (1 - P_{i,i+1}) \qquad \text{on} \qquad \mathcal{H} = \left(\mathbb{C}^2\right)^{\otimes L}, \tag{9}$$

where $P_{ij} = (1 + \vec{\sigma}_i \cdot \vec{\sigma}_j)/2$ is the permutation operator for the spins at sites $i$ and $j$. It commutes with all of the global $\mathfrak{sl}_2$, which acts on the spin chain by

$$S^{\pm} = \sum_{i=1}^{L} \sigma_i^{\pm}, \qquad S^z = \frac{1}{2}\sum_{i=1}^{L} \sigma_i^z,$$
$$[S^z, S^{\pm}] = \pm S^{\pm}, \qquad [S^+, S^-] = 2S^z. \tag{10}$$

We denote the coordinate basis vectors of $\mathcal{H}$ by

$$|i_1, \ldots, i_M\rangle\rangle \equiv \sigma_{i_1}^- \cdots \sigma_{i_M}^- |\uparrow \cdots \uparrow\rangle. \tag{11}$$

Bethe's exact characterisation of the spectrum of (9) [44] is one of the cornerstones of integrability. It admits an algebraic reformulation in the framework of the quantum inverse-scattering method developed by Faddeev *et al* [45]. One of the many benefits of this framework is that it allows for the construction of *inhomogeneous* generalisations of (9) depending on inhomogeneity parameters $\theta_1, \ldots, \theta_L$ that break translational invariance (homogeneity) without spoiling integrability. Let us briefly review how this goes. We start from the rational $R$-matrix [46, 47]

$$\bar{R}(u) = u + \mathrm{i}P, \qquad \bar{R}(u - v) = \begin{array}{c} v \\ \longleftarrow\!\!\!\!|\!\!\longrightarrow u \\ \downarrow \end{array}, \tag{12}$$

acting on $\mathbb{C}^2 \otimes \mathbb{C}^2$. Here and below we use a bar to distinguish the standard conventions for Heisenberg from those that we will use for the spin-Calogero–Sutherland model in §3.

Introducing another copy of $\mathbb{C}^2$ as an 'auxiliary' space (which we label by 0) the monodromy matrix on $\mathbb{C}^2 \otimes \mathcal{H}$ is defined as[3]

$$\overline{T}_0(u) = \overline{R}_{01}(u - \theta_1 - \mathrm{i}/2) \cdots \overline{R}_{0L}(u - \theta_L - \mathrm{i}/2) = \quad \text{(13)}$$

where the subscripts indicate the spaces in which the $R$-matrices act, and we included an inhomogeneity $\theta_i$ at each site. We can write $T_0$ as a $2 \times 2$ matrix whose elements act on the physical space $\mathcal{H}$,

$$\overline{T}_0(u) = \begin{pmatrix} \overline{A}(u) & \overline{B}(u) \\ \overline{C}(u) & \overline{D}(u) \end{pmatrix}_0 . \tag{14}$$

The *twisted* transfer matrix with twist $\kappa \in \mathbb{C}$ is a twisted trace over the auxiliary space,

$$\overline{t}(u;\kappa) = \mathrm{Tr}_0\!\left[ \kappa^{\sigma_0^z} \, \overline{T}_0(u) \right] = \kappa\, \overline{A}(u) + \kappa^{-1} \overline{D}(u) = \quad \text{(15)}$$

where we focus on a diagonal twist

$$\kappa^{\sigma^z} = \begin{pmatrix} \kappa & 0 \\ 0 & \kappa^{-1} \end{pmatrix} = \frac{\kappa + \kappa^{-1}}{2} + \frac{\kappa - \kappa^{-1}}{2}\, \sigma^z . \tag{16}$$

This operator acts on $\mathcal{H}$. Note that $\kappa \neq \pm 1$ breaks the global $\mathfrak{sl}_2$-symmetry down to (its Cartan subalgebra) $S^z$. If we view the twist as a formal parameter we further have a discrete symmetry (Weyl group of $\mathfrak{sl}_2$) that flips all $\uparrow \leftrightarrow \downarrow$ and inverts the twist $\kappa \to \kappa^{-1}$. The transfer matrix provides a family of commuting operators,

$$\left[ \overline{t}(u;\kappa), \overline{t}(v;\kappa) \right] = 0 , \tag{17}$$

that can be diagonalised simultaneously. The commutative algebra generated by the transfer matrix is maximal for generic twist [7–9], and is called the *Bethe algebra*. We give some of its explicit elements in Sections 2.3.1–2.3.2.

## 2.2 Algebraic Bethe Ansatz and QQ-relation

The aim of the algebraic Bethe Ansatz is to construct the eigenvectors of the transfer matrix using the Yangian generators (14). Let us first review the case when the inhomogeneities $\theta_i$ and the twist $\kappa$ are in generic position. As a starting point we take a reference or '(pseudo)vacuum' state $|0\rangle$, annihilated by the $C$-operator from (14), $\overline{C}(u)|0\rangle = 0$. Later we will encounter more complicated (pseudo)vacua, but for now we have

$$|0\rangle = |\uparrow\uparrow \cdots \uparrow\rangle , \tag{18}$$

which is an eigenstate of $\overline{t}(u;q)$ since the latter preserves the number of $\downarrow$s. To build the other states, we act on it with the $B$-operator, which serves as a creation operator,

$$\overline{B}(u_1) \cdots \overline{B}(u_M)|0\rangle . \tag{19}$$

---

[3]Like the i in (12), the shifts in (13) ensure that the Bethe equations will come out in their usual form. Also note that one may incorporate the twist from (15) in the monodromy matrix instead. As we consider diagonal twist, this does not change the commutation relations for the operators (14). We choose to include the twist in (15) so that (14) are independent of the twist, which will be convenient when we take limits of the twist.

This is an eigenstate of the transfer matrix with eigenvalue

$$\bar{\tau}(u;\kappa) = \kappa \prod_{i=1}^{L}(u-\theta_i+\mathrm{i}/2)\prod_{m=1}^{M}\frac{u-u_m-\mathrm{i}}{u-u_m} + \kappa^{-1}\prod_{i=1}^{L}(u-\theta_i-\mathrm{i}/2)\prod_{m=1}^{M}\frac{u-u_m+\mathrm{i}}{u-u_m}\,, \qquad (20)$$

provided the parameters $u_m$, known as Bethe roots, satisfy the Bethe-Ansatz equations

$$\kappa^2 \prod_{i=1}^{L}\frac{u_m-\theta_i+\mathrm{i}/2}{u_m-\theta_i-\mathrm{i}/2} = \prod_{n(\neq m)}^{M}\frac{u_m-u_n+\mathrm{i}}{u_m-u_n-\mathrm{i}}\,, \qquad 1\leqslant m\leqslant M\,. \qquad (21)$$

Note that these equations ensure that (20) is a polynomial of degree $L$ in $u$, in accordance with the definition (13) of the monodromy matrix, whose coefficients depend on the Bethe roots $u_m$ as well as the inhomogeneities $\theta_i$ and twist $\kappa$. Also note that the Bethe vectors only depend on the twist through the Bethe roots. Further observe that the Bethe equations are *symmetric* in the inhomogeneities. For generic values of the parameters, i.e. $\theta_i \neq \theta_j + \mathrm{i}$ for all $(i,j)$ and $\kappa \notin \{0,\pm 1,\infty\}$, one should take all admissable solutions $\{u_1,\ldots,u_M\}$ of (21) for $M \in \{0,1,\ldots,L\}$. Here, *admissible* means that the solution does not contain any coincident Bethe roots, i.e. $u_m \neq u_n$ for all $m \neq n$. It is known that there are precisely $\binom{L}{M}$ distinct solutions [7–9], accounting for all $M$-magnons eigenstates of the transfer matrix via the algebraic Bethe Ansatz (19).

The Bethe equations admit several reformulations. We will use the standard shorthand

$$f^{\pm}(u) \equiv f(u\pm\mathrm{i}/2)\,, \qquad f^{\pm\pm} \equiv (f^{\pm})^{\pm}\,, \qquad (22)$$

and define the polynomial

$$\bar{Q}_{\boldsymbol{\theta}}(u) \equiv \prod_{i=1}^{L}(u-\theta_i)\,, \qquad (23)$$

so that $\bar{Q}_{\boldsymbol{\theta}}^{\pm}$ are the eigenvalues of $\bar{A}$ and $\bar{D}$ on $|0\rangle$, respectively. Further introduce Baxter's *Q-function* as the polynomial whose zeroes are the Bethe roots:

$$\bar{Q}(u) \equiv \prod_{m=1}^{M}(u-u_m)\,. \qquad (24)$$

The transfer-matrix eigenvalue (20) then takes the concise form

$$\bar{\tau}(u;\kappa) = \kappa\,\bar{Q}_{\boldsymbol{\theta}}^{+}\,\frac{\bar{Q}^{--}}{\bar{Q}} + \kappa^{-1}\,\bar{Q}_{\boldsymbol{\theta}}^{-}\,\frac{\bar{Q}^{++}}{\bar{Q}}\,, \qquad (25)$$

and the Bethe equations (21) read

$$\kappa^2\,\frac{\bar{Q}_{\boldsymbol{\theta}}^{+}}{\bar{Q}_{\boldsymbol{\theta}}^{-}} = -\frac{\bar{Q}^{++}}{\bar{Q}^{--}}\,, \quad \text{at} \quad u=u_m\,, \qquad 1\leqslant m\leqslant M\,. \qquad (26)$$

A particularly convenient reformulation of the latter is as follows. Defined the counterpart of (24) 'beyond the equator', of degree $L-M$, as

$$\widetilde{\bar{Q}}(u) \equiv \prod_{n=1}^{L-M}(u-v_n)\,. \qquad (27)$$

For spin-1/2 the *Wronskian Bethe equations* are a functional equation also called the $\mathfrak{sl}_2$ *QQ-relation*, see [50,83] and references therein,

$$\left(\kappa-\kappa^{-1}\right)\bar{Q}_{\boldsymbol{\theta}} = \kappa\,\bar{Q}^{-}\,\widetilde{\bar{Q}}^{+} - \kappa^{-1}\,\bar{Q}^{+}\,\widetilde{\bar{Q}}^{-}\,. \qquad (28)$$

Demanding that $\bar{Q}$ and $\widetilde{\bar{Q}}$ be of the form (24) and (27) and solving (28) yields a discrete set of solutions for the Bethe roots. One can show from (28) that the roots $u_m$ of $\bar{Q}$ satisfy the Bethe equations, see Appendix B.1.[4] (For the periodic case with $\kappa = 1$, see Section 2.3.3.)

While for generic values of $\theta_i, \kappa$ the Bethe equations and QQ-relation are equivalent, this is not always the case, cf. [48–51]. The QQ-relation is often more useful as its solutions are in bijection with the transfer-matrix eigenstates. This *completeness* of the Bethe Ansatz was proved for almost all values of the inhomogeneities (including the homogeneous limit with all $\theta_i = 0$) [7–9].

## 2.3 Commuting charges

### 2.3.1 Inhomogeneous analogues of translation operator

One way to calculate elements of the Bethe algebra is the inhomogeneous analogue of the standard approach: evaluating the transfer matrix and its logarithmic derivative(s) at a special point $u_*$ where at least one of the $R$-matrices in the product (13) simplifies. Since $\bar{R}(0) = \mathrm{i}P$ any $u_* = \theta_j + \mathrm{i}/2$ will do the job. We compute (as in Section 2.1, 'H' is for 'Heisenberg')

$$
\begin{aligned}
G_j^{\mathrm{H}}(\kappa) &\equiv -\mathrm{i}\,\bar{t}(\theta_j + \mathrm{i}/2; \kappa) \\
&= -\mathrm{i}\,\mathrm{Tr}_0\big[\kappa^{\sigma_0^z}\bar{R}_{01}(\theta_j - \theta_1)\cdots\bar{R}_{0L}(\theta_j - \theta_L)\big] = \\
&= \bar{R}_{j,j+1}(\theta_j - \theta_{j+1})\cdots\bar{R}_{jL}(\theta_j - \theta_L)\kappa^{\sigma_j^z}\bar{R}_{j1}(\theta_j - \theta_1)\cdots\bar{R}_{j,j-1}(\theta_j - \theta_{j-1}).
\end{aligned}
\tag{29}
$$

These operators, which are sometimes called 'scattering operators' [11, 47, 52, 53], obey the relations

$$
[G_i^{\mathrm{H}}(\kappa), G_j^{\mathrm{H}}(\kappa)] = 0, \qquad \prod_{j=1}^{L}G_j^{\mathrm{H}}(\kappa) = \prod_{j=1}^{L}\kappa^{\sigma_j^z} = \kappa^{2S^z}.
\tag{30}
$$

In the semiclassical limit these operators reduce to the Hamiltonians of the rational Gaudin model [84]. In the homogeneous limit all (29) reduce to the (twisted) translation operator

$$
\theta_i \to \theta: \qquad G_j^{\mathrm{H}}(\kappa) \to G^{\mathrm{H}}(\kappa) = \kappa^{\sigma_1^z}P_{12}\cdots P_{L-1,L}, \qquad G^{\mathrm{H}}(\kappa)^L = \kappa^{2S^z}.
\tag{31}
$$

The eigenvalues of the $G_i(\kappa)$ are conserved, and for $\theta_i \to \theta$ give rise to the notion of (twisted) momentum on the lattice, see e.g. §B.2 of [50]. For general inhomogeneities, however, there is no single analogue of the momentum.

One obtains $L$ inhomogeneous deformations of the spin-chain Hamiltonian (9) via the logarithmic derivative $H_j^{\mathrm{H}}(\kappa) \equiv G_j^{\mathrm{H}}(\kappa)^{-1}\,\partial_u|_{u=\theta_j+\mathrm{i}/2}\,\bar{t}(u; \kappa)$. The spin chain given by these commuting operators has long-range interactions involving multiple spins at a time, with terms that resemble the interactions of the $q$-deformed (XXZ-type) Haldane–Shastry spin chain [54, 55]. For our purposes, however, they are not suitable.[5]

### 2.3.2 Another family of conserved charges with inhomogeneities

Let us instead expand the transfer matrix at $u = 0$ or $u \to \infty$. We pick the latter option and expand it as a (formal) power series in $(-\mathrm{i}u)^{-1}$. In order to remove some trivial contributions to the charges, we find it convenient to expand (cf. the shifts in (13))

$$
\frac{\bar{t}(u + \mathrm{i}/2; \kappa)}{\bar{Q}_\theta(u)} = \kappa + \kappa^{-1} + \sum_{n=1}^{\infty}\bar{t}_n(\kappa)(-\mathrm{i}u)^{-n}.
\tag{32}
$$

---

[4]Likewise, the 'dual Bethe roots' $v_n$ of $\widetilde{\bar{Q}}$ satisfy the Bethe equations 'beyond equator', with $M \rightsquigarrow L-M$, $\kappa \rightsquigarrow \kappa^{-1}$.

[5]The values $u_* = \theta_j + \mathrm{i}/2$ break the symmetry between the inhomogeneities, and are not compatible with the fermionic condition $s_{ij}P_{ij} = -1$ ($i \neq j$) from Section 3.2.

Here $\kappa + \kappa^{-1}$ is the quantum dimension of the auxiliary space. The next few coefficients are

$$\bar{t}_1(\kappa) = \sum_{i=1}^{L} \kappa^{\sigma_i^z} = \frac{\kappa + \kappa^{-1}}{2} L + (\kappa - \kappa^{-1}) S^z,$$

$$\bar{t}_2(\kappa) = \sum_{i<j} \kappa^{\sigma_j^z} P_{ij} - i \sum_{i=1}^{L} \theta_i \, \kappa^{\sigma_i^z}, \tag{33}$$

$$\bar{t}_3(\kappa) = \sum_{i<j<k} \kappa^{\sigma_k^z} P_{jk} P_{ij} - i \sum_{i<j} (\theta_i + \theta_j) \kappa^{\sigma_j^z} P_{ij} - \sum_{i=1}^{L} \theta_i^2 \, \kappa^{\sigma_i^z}.$$

Here and below, by $\sum_{i<j}$ we mean the sum over all $1 \leqslant i < j \leqslant L$, and similarly for $\sum_{i<j<k}$.

The eigenvalues of (33) are obtained by analogously expanding (25):

$$\frac{\bar{\tau}(u + i/2; \kappa)}{\bar{Q}_\theta(u)} = \kappa + \kappa^{-1} + \sum_{n=1}^{\infty} \bar{\tau}_n(\kappa) (-iu)^{-n}. \tag{34}$$

The first two coefficients read

$$\bar{\tau}_1(\kappa) = \frac{\kappa + \kappa^{-1}}{2} L + (\kappa - \kappa^{-1}) \left( \frac{L}{2} - M \right), \tag{35}$$

$$\bar{\tau}_2(\kappa) = \frac{\kappa + \kappa^{-1}}{2} \bar{\tau}_2(1) + \frac{\kappa - \kappa^{-1}}{2} \left[ -i \sum_{i=1}^{L} \theta_i + 2i \sum_{m=1}^{M} u_m + (L-1) \left( \frac{L}{2} - M \right) \right], \tag{36}$$

where in the untwisted case

$$\bar{\tau}_2(1) = -i \sum_{i=1}^{L} \theta_i + \left( \frac{L}{2} - M \right) \left( \frac{L}{2} - M + 1 \right) + \frac{1}{4} L(L-4). \tag{37}$$

### 2.3.3 Periodic case

Removing the twist by setting $\kappa = 1$ enhances the $S^z$-symmetry of the transfer matrix $\bar{t}(u; \kappa)$ to global $\mathfrak{sl}_2$ symmetry under (10). Its eigenspaces become degenerate, forming spin multiplets (irreducible $\mathfrak{sl}_2$-representations) with the same eigenvalue of $\bar{t}(u)$. As a matter of fact, $\bar{t}_2(1)$ is essentially the quadratic Casimir

$$\vec{S} \cdot \vec{S} = \frac{1}{2} (S^+ S^- + S^- S^+) + S^z S^z = \sum_{i<j} P_{ij} - \frac{1}{4} L(L-4), \tag{38}$$

where the constant $-\frac{1}{4} L(L-4) = \frac{L}{2} \left( \frac{L}{2} + 1 \right) - \frac{1}{2} L(L-1)$ accounts for the difference in eigenvalues on $|\uparrow \cdots \uparrow\rangle$. The first non-trivial charge is then $\bar{t}_3(1)$, whose eigenvalue is

$$\bar{\tau}_3(1) = -\sum_{i=1}^{L} \theta_i^2 - i(L - M - 1) \sum_{i=1}^{L} \theta_i + 2i \left( \frac{L}{2} - M + 1 \right) \sum_{m=1}^{M} u_m$$

$$+ \frac{1}{3} \left( \frac{L}{2} - M - 1 \right) [L(L - M - 1) + M(M - 1)]. \tag{39}$$

As long as the inhomogeneities are generic (including the homogeneous case) the Bethe-Ansatz construction (19) of the eigenstates provides the highest-weight vector in each multiplet, by solving the Wronskian Bethe equations up to the equator $M \leqslant \lfloor L/2 \rfloor$.[6] The descendants

---

[6] At $\kappa = 1$ subtleties occur that require care. For singular solutions, containing exact strings (i.e. $u_m - u_n = i$), the eigenvector and eigenvalues need to be regularised. This already happens at $M = 2$ when $L$ is even.

in the multiplet can be obtained from it by acting with the global lowering operator $S^-$. This fits in the framework of the algebraic Bethe Ansatz: the leading term of the expansion of $\bar{B}(u)$ in $u$ occurs at order $L-1$ with coefficient $S^-$ up to a constant, so acting with $S^-$ can be viewed as adding a magnon with an infinite Bethe root. Indeed, when $\kappa = 1$ then from any solution to the Bethe equations with given $M$ one formally obtains another solution to the Bethe equations by adding $u_{M+1} = \infty$ to the solution. In the absence of a twist, the Q-functions are still monic polynomials but the degree of $\widetilde{\bar{Q}}$ is now $L-M+1$ instead of $L-M$, and the factor $\kappa - \kappa^{-1}$ in the QQ-relation (28) is replaced by $L-2M+1$ [85, 86].

### 2.3.4 Extreme twist and Gelfand–Tsetlin basis

For an extreme twist $\kappa \to \infty$ the twisted transfer matrix simplifies to $\bar{t}(u; \kappa) \sim \kappa \bar{A}(u)$. Of the global $\mathfrak{sl}_2$ only $S^z$ remains as a symmetry. Together with the quantum determinant (which is independent of the twist), the $A$-operator generates a subalgebra of the Yangian called the *Gelfand–Tsetlin subalgebra*, and the Bethe vectors reduce to the *Gelfand–Tsetlin basis* for the spin chain [56, 57].[7] This limit provides a useful combinatorial model for the Yangian representation. The QQ-relation (28) in this case takes the factorised form

$$\bar{Q}_{\boldsymbol{\theta}}(u) = \prod_{m=1}^{M}\left(u - u_m - \frac{\mathrm{i}}{2}\right)\prod_{n=1}^{L-M}\left(u - v_n + \frac{\mathrm{i}}{2}\right), \tag{40}$$

with the left-hand side given by (23). Comparing zeroes gives explicit values for the Bethe roots, sticking to the inhomogeneities as $u_m = \theta_{i_m} - \mathrm{i}/2$ for $I = \{i_1, \ldots, i_M\} \subset \{1, \ldots, L\}$ in the $M$-magnon sector, and the $2^L$ spin-chain states are given by all possible choices of such subsets $I$. The corresponding eigenvalues of the transfer matrix $\bar{A}(u)$ factorise as well,

$$\bar{\alpha}_I(u) = \prod_{i \in I}\left(u - \theta_i - \frac{\mathrm{i}}{2}\right)\prod_{j \notin I}\left(u - \theta_j + \frac{\mathrm{i}}{2}\right). \tag{41}$$

Note that the Bethe roots do not 'interact': from a solution $\{\theta_i - \mathrm{i}/2\}_{i \in I}$ we get another solution by just adding any $\theta_j - \mathrm{i}/2$ with $j \notin I$. Moreover, the corresponding eigenvectors are simply related by acting with $\bar{B}(\theta_j - \mathrm{i}/2)$, which affects the eigenvalue of the $A$-operator by changing the factor $u - \theta_j + \mathrm{i}/2$ in (41) to $u - \theta_j - \mathrm{i}/2$. This should be contrasted with the usual situation at finite twist, where adding a root to a solution affects all other roots (except for infinite roots, corresponding to descendants, at $\kappa = \pm 1$), and one *first* has to construct the 'off-shell' Bethe vector $\bar{B}(u_1)\cdots\bar{B}(u_M)|0\rangle$ and *then* plug in a solution to get a transfer-matrix eigenvector 'on-shell'. Let us also point out that the Bethe states in this case coincide, up to normalisation, with the vectors of Sklyanin's separation-of-variables (SoV) basis for the Heisenberg spin chain with anti-periodic boundary conditions, see equation (3.10) in [58]. The Yangian Gelfand–Tsetlin basis also plays a central role in the SoV approach for more general twist and higher rank [59].

## 2.4 Fusion

Now we turn to the role of inhomogeneities and fusion of Yangian representations, developed originally by Kulish, Reshetikhin and Sklyanin [60].

---

[7]If $\kappa \to 0$ then $\bar{t}(u; \kappa) \sim \kappa^{-1}\bar{D}(u)$ yields another Gelfand–Tsetlin subalgebra, and Bethe roots $u_m = \theta_{i_m} + \mathrm{i}/2$. Physically, the twist is $\kappa = \mathrm{e}^{\mathrm{i}\varphi/2}$, so these limits correspond to extreme *imaginary* twists.

### 2.4.1 General description

While the monodromy matrix and its four operator entries depend on the ordering of the inhomogeneities $\theta_i$, as long as the inhomogeneities are generic they can be reordered between the sites of the spin chain via a similarity transformation on the Hilbert space. To see this, note that the Yang-Baxter equation

$$\bar{R}_{0i}(u-v)\bar{R}_{0j}(u-w)\bar{R}_{ij}(v-w) = \bar{R}_{ij}(v-w)\bar{R}_{0j}(u-w)\bar{R}_{0i}(u-v)\,, \tag{42}$$

implies that the operator

$$\check{\bar{R}}(u) \equiv P\,\bar{R}(u) = \mathrm{i} + u\,P\,, \qquad \begin{matrix} v & u \\ \diagdown\!\!\!\diagup \end{matrix} = \check{\bar{R}}(u-v)\,, \tag{43}$$

permutes inhomogeneities. (Note that for this operator the subscripts of the spaces do not follow the lines, unlike for the $R$-matrix (12). The parameters follow the lines in both cases.) Indeed, if we multiply by $P_{ij}$ from the left and take $i = j+1$ and $v = \theta_{j+1} + \mathrm{i}/2, w = \theta_j + \mathrm{i}/2$, we obtain the intertwining relation

$$\begin{aligned} &\bar{R}_{0j}(u-\theta_{j+1}-\mathrm{i}/2)\bar{R}_{0,j+1}(u-\theta_j-\mathrm{i}/2)\check{\bar{R}}_{j+1,j}(\theta_{j+1}-\theta_j) \\ &= \check{\bar{R}}_{j+1,j}(\theta_{j+1}-\theta_j)\bar{R}_{0j}(u-\theta_j-\mathrm{i}/2)\bar{R}_{0,j+1}(u-\theta_{j+1}-\mathrm{i}/2)\,. \end{aligned} \tag{44}$$

This extends to the monodromy matrix (13),[8]

$$\bar{T}_0(u;\dots,\theta_{j+1},\theta_j,\dots)\check{\bar{R}}_{j+1,j}(\theta_{j+1}-\theta_j) = \check{\bar{R}}_{j+1,j}(\theta_{j+1}-\theta_j)\bar{T}_0(u;\dots,\theta_j,\theta_{j+1},\dots)\,, \tag{45a}$$

that is,

$$\text{(graphical diagram)} \qquad u-\mathrm{i}/2\,. \quad = \quad \text{(graphical diagram)} \qquad u-\mathrm{i}/2\,. \tag{45b}$$

This will be the central identity in what follows. There are two scenarios to consider: the generic, irreducible case, and (two) cases with an invariant subspace leading to fusion.

Note that

$$\det\check{\bar{R}}(u) = -(u+\mathrm{i})^3\,(u-\mathrm{i})\,. \tag{46}$$

As long as $\check{\bar{R}}_{j+1,j+1}(\theta_j - \theta_{j+1})$ is invertible, i.e. $\theta_j - \theta_{j+1} \neq \pm\mathrm{i}$, (45) gives

$$\bar{T}_0(u;\dots,\theta_{j+1},\theta_j,\dots) = \check{\bar{R}}_{j+1,j}(\theta_{j+1}-\theta_j)\,\bar{T}_0(u;\dots,\theta_j,\theta_{j+1},\dots)\check{\bar{R}}_{j+1,j}(\theta_{j+1}-\theta_j)^{-1}\,. \tag{47}$$

Thus, any two inhomogeneities can be exchanged by a similarity transformation on $\mathcal{H}$ consisting of a sequence of conjugations by $R$-matrices that are all invertible provided $\theta_i - \theta_j \neq \pm\mathrm{i}$ for all $i, j$. This property is inherited by the transfer matrix $\bar{t}(u)$, whose spectrum is independent of the order of the inhomogeneities. This is reflected in the symmetry in the $\theta_i$ of the Bethe equations (21) and Baxter equation (25). Here the algebraic Bethe Ansatz can be used to construct the whole spectrum from $|0\rangle = |\uparrow\cdots\uparrow\rangle$ (Figure 2). This includes the homogeneous limit where all $\theta_i = 0$ are equal, yielding the ordinary Heisenberg XXX spin chain (9).

---

[8]By the symmetry $\bar{R}_{ji}(u) \equiv P_{ij}\bar{R}_{ij}(u)P_{ij} = \bar{R}_{ij}(u)$ the order of the subscripts of the $R$-matrix in (45) does not matter. The decreasing subscripts are due to the order of the physical spaces in (13), cf. the graphical notation.

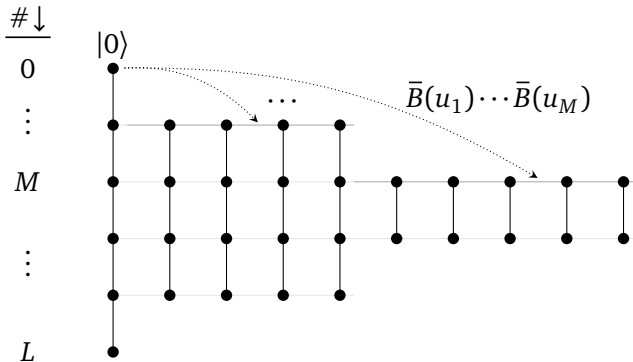

Figure 2: Structure of $\mathcal{H}$ for generic inhomogeneities ($\theta_i - \theta_j \neq \pm\mathrm{i}$ for all $i, j$). Each dot represents an eigenstate, organised into $\mathfrak{sl}_2$-irreps $V_d$ as shown by the black vertical lines, with vertical axis recording $M = L/2 - S^z$. The dotted lines indicate (the algebraic Bethe Ansatz) part of the Yangian action. The 'off shell' Bethe vectors $\bar{B}(u_1)\cdots\bar{B}(u_M)|0\rangle$ span the entire $M$-particle sector, as sketched by the gray horizontal lines (lighter for the $\mathfrak{sl}_2$-descendants, obtained by acting with $S^- \sim \bar{B}(\infty)$ in the periodic case). The Bethe equations for $u_1, \ldots, u_M$ single out the points in these subspaces that are eigenvectors of the transfer matrix.

In terms of representation theory, an inhomogeneity $\theta_i$ is the parameter of an evaluation representation of the Yangian, and the spin-chain Hilbert space is a tensor product of such evaluation representations. For generic values of the inhomogeneities ($\theta_i - \theta_j \neq \pm\mathrm{i}$ for all $i, j$) the Hilbert space is *irreducible*, and the Yangian representation is called *tame*. In this case, (47) says that $\check{\bar{R}}_{j,j+1}(\theta_j - \theta_{j+1})$ intertwines the Yangian irreps on $\mathcal{H}$ with inhomogeneities $(\ldots, \theta_j, \theta_{j+1}, \ldots)$ and $(\ldots, \theta_{j+1}, \theta_j, \ldots)$. The completeness of the Bethe Ansatz was proved for even more general values of the inhomogeneities (namely whenever $\theta_i - \theta_j \neq \mathrm{i}$ for all $i < j$.) in [7,8], see also [9]. We remark that the exchange relation (47) also appears as the 'local condition' of the Knizhnik–Zamolodchikov (KZ) system [52,61].

If the $R$-matrix in (45) is not invertible we cannot write (47), but instead there is an *invariant* subspace. This can be seen as follows. When the two inhomogeneities differ by $\pm\mathrm{i}$, the $R$-matrix (43) becomes proportional to an (anti)symmetriser,

$$\check{R}(\pm\mathrm{i}) = 2\,\mathrm{i}\,\Pi^\pm, \qquad \Pi^\pm = (1 \pm P)/2. \tag{48}$$

Let us write $V_d$ for the spin-$s$ irrep of $\mathfrak{sl}_2$, which has dimension $d = 2s + 1$. The operators (48) are orthogonal projectors decomposing two spin-1/2 sites into a triplet and singlet,

$$V_2 \otimes V_2 \supset \Pi^+(V_2 \otimes V_2) \cong V_3, \qquad V_2 \otimes V_2 \supset \Pi^-(V_2 \otimes V_2) \cong V_1, \tag{49}$$

corresponding to the (Clebsch–Gordan) decomposition

$$V_2 \otimes V_2 \cong V_3 \oplus V_1, \qquad \text{for } \mathfrak{sl}_2. \tag{50}$$

Focussing on sites $j$ and $j+1$ of the Hilbert space, this decomposition gives two orthogonal subspaces of $\mathcal{H}$ of dimension $3 \times 2^{L-2}$ and $2^{L-2}$, respectively:

$$\begin{aligned}
\mathcal{H} = V_2^{\otimes L} &= \Pi^+_{j,j+1}(\mathcal{H}) \oplus \Pi^-_{j,j+1}(\mathcal{H}) \\
&\cong \left(V_2^{\otimes(j-1)} \otimes V_3 \otimes V_2^{\otimes(L-j-1)}\right) \oplus \left(V_2^{\otimes(j-1)} \otimes V_1 \otimes V_2^{\otimes(L-j-1)}\right), \qquad \text{for } \mathfrak{sl}_2.
\end{aligned} \tag{51}$$

While these two subspaces are generically mixed by the monodromy matrix, in special cases one of them is preserved. To see this we return to the relation (45). When $\theta_{j+1} = \theta_j \mp \mathrm{i}$, the left-hand side of (45) annihilates any vector in $\ker \Pi^{\mp}_{j,j+1}$. Yet on the right-hand side the projector acts after the monodromy matrix, so (45) implies that $\overline{T}_0(u; \theta_1, \ldots, \theta_j, \theta_j \mp \mathrm{i}, \ldots, \theta_L)$ must preserve $\ker \Pi^{\mp}_{j,j+1} = \Pi^{\pm}_{j,j+1}(\mathcal{H})$. Given a choice of inhomogeneities with an adjacent pair differing by $\mp \mathrm{i}$ we can thus restrict the monodromy matrix to the subspace $\Pi^{\pm}_{j,j+1}(\mathcal{H})$ to get a copy of an inhomogeneous spin chain of length $L-1$, containing $L-2$ sites of spin $1/2$ plus a spin 1 (triplet) or 0 (singlet) at site $j$, cf. (51). In Appendix A we show that the factors $\overline{R}_{0j}(u - \theta_j - \mathrm{i}/2)\overline{R}_{0,j+1}(u - \theta_{j+1} - \mathrm{i}/2)$ from the monodromy matrix yield a single $R$-matrix acting on (the spin-$1/2$ auxiliary space and) site $j$ with spin 1 or 0. This construction is called *fusion* [60].

There is no reason for the monodromy matrix $\overline{T}_0(u; \theta_1, \ldots, \theta_j, \theta_j \mp \mathrm{i}, \ldots, \theta_L)$ to preserve the complementary space $\Pi^{\mp}_{j,j+1}(\mathcal{H})$ as well—and indeed it does not, as we will illustrate shortly. In terms of the transfer matrix, general complex values of inhomogeneities spoil hermiticity, so its eigenspaces are not orthogonal.

In terms of representation theory, the preceding says that if $\theta_{j+1} - \theta_j = \mp \mathrm{i}$ then the Yangian representation on $\mathcal{H}$ with inhomogeneities $\theta_1, \ldots, \theta_j, \theta_{j+1}, \ldots, \theta_L$ is *reducible*. However, since the orthogonal complement is not preserved by the Yangian, this reducible representation is *indecomposable*.[9] In Appendix D.1 we illustrate what this means concretely for $L = 2$. Since the dimension of the invariant subspace depends on the sign, the Yangian representations on $\mathcal{H}$ with inhomogeneities $(\theta_1, \ldots, \theta_j, \theta_j + \mathrm{i}, \ldots, \ldots, \theta_L)$ versus $(\theta_1, \ldots, \theta_j + i, \theta_j, \ldots, \ldots, \theta_L)$ are *not* isomorphic.

So how does all of this affect us in practice when we want to use the Bethe Ansatz for an inhomogeneous Heisenberg XXX spin chain? In the next two subsections we illustrate this, partially based on numerics for examples with small length $L$.

### 2.4.2 Bethe Ansatz for fusion into singlet

Consider first the case where two sites are fused into a singlet. The inhomogeneities are generic, except for one pair that we may take to be adjacent using the intertwiners,

$$\theta_{j+1} = \theta_j + \mathrm{i}. \tag{53}$$

This is precisely the non-generic case for which completeness holds [9]. Considering the periodic case $\kappa = 1$ for simplicity, we have the following features, as is illustrated in Appendix D for examples with low $L$.

- As we have just discussed, fixing the singlet state $(|\uparrow\downarrow\rangle - |\downarrow\uparrow\rangle)/\sqrt{2}$ at sites $j$ and $j+1$ and allowing any spin configuration at the $L-2$ remaining sites together form a Yangian-invariant subspace of $\mathcal{H}$ of dimension $2^{L-2}$,

$$V_{\mathrm{inv}} = \Pi^{-}_{j,j+1}(\mathcal{H}) \cong V_2^{\otimes(j-1)} \otimes V_1 \otimes V_2^{\otimes(L-j-1)}. \tag{54}$$

---

[9]Another situation where reducible but indecomposable representations appear is for $U_q(\mathfrak{sl}_2)$ with $q$ a root of unity, see e.g. [62]. In the mathematical literature this situation is often described via non-split short exact sequences. In brief, the coimage $\mathrm{coim}(\Pi^{\mp}_{j,j+1}) \equiv \mathcal{H}/\ker(\Pi^{\mp}_{j,j+1})$ by definition fits in the short exact sequence

$$0 \longrightarrow \ker(\Pi^{\mp}_{j,j+1}) \longrightarrow \mathcal{H} \longrightarrow \mathrm{coim}(\Pi^{\mp}_{j,j+1}) \longrightarrow 0, \tag{52}$$

where *exactness* means that the image of each map is the kernel of the next one. As $\mathfrak{sl}_2$-modules, this sequence *splits* by (51). This is closely related to the fact that $\mathrm{coim}(\Pi^{\mp}_{j,j+1}) \cong \mathrm{coker}(\Pi^{\pm}_{j,j+1}) \cong \ker(\Pi^{\pm}_{j,j+1}) \cong \mathrm{im}(\Pi^{\mp}_{j,j+1})$ as $\mathfrak{sl}_2$-modules. For the Yangian, acting by $\overline{T}_0(u; \theta_1, \ldots, \theta_j, \theta_j \mp \mathrm{i}, \ldots, \theta_L)$, the sequence (52) remains exact: $V_{\mathrm{inv}} = \ker(\Pi^{\mp}_{j,j+1}) \subset \mathcal{H}$ is a Yangian submodule, and the quotient $\mathrm{coim}(\Pi^{\mp}_{j,j+1})$ is also a Yangian module. However, this time (52) is not split: $\ker(\Pi^{\mp}_{j,j+1}) \oplus \mathrm{coim}(\Pi^{\mp}_{j,j+1})$ is *not* isomorphic to $\mathcal{H}$ as a $Y\mathfrak{gl}_2$-module. The above equivalence between $\mathrm{coim}(\Pi^{\mp}_{j,j+1})$ and $\mathrm{im}(\Pi^{\mp}_{j,j+1})$ does not respect the Yangian, and $\mathrm{im}(\Pi^{\mp}_{j,j+1})$ is not even a $Y\mathfrak{gl}_2$-module.

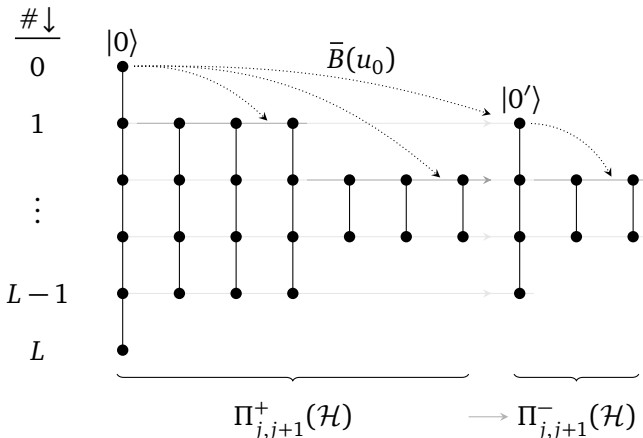

Figure 3: Structure of $\mathcal{H}$ (cf. Figure 2) with $\theta_{j+1} = \theta_j + i$ and other $\theta_i$ generic. As indicated, the Yangian action may send vectors in $\Pi^+_{j,j+1}(\mathcal{H})$, like $|0\rangle = |\uparrow \cdots \uparrow\rangle$, to anywhere in $\mathcal{H}$, but cannot get out of $V_{\text{inv}} = \Pi^-_{j,j+1}(\mathcal{H})$. In particular, $|0'\rangle = \bar{B}(u_0)|0\rangle$ with $u_0 = \theta_j + i/2$ is a good reference state for the algebraic Bethe Ansatz inside $V_{\text{inv}}$.

On this invariant subspace, $\bar{T}_0(u; \theta_1, \ldots, \theta_j, \theta_j + i, \ldots, \theta_L)$ looks just like the monodromy matrix for an inhomogeneous spin chain with 'effective length' $L - 2$.

- The Bethe Ansatz is complete in the sense that *all* eigenstates of the transfer matrix are given by the algebraic Bethe Ansatz (19). It works as follows.

- Consider the Bethe equations (21). Due to (53) one numerator and denominator cancel on the left-hand side, yielding

$$\frac{u_m - \theta_j + i/2}{u_m - \theta_j - 3i/2} \prod_{i(\neq j, j+1)}^{L} \frac{u_m - \theta_i + i/2}{u_m - \theta_i - i/2} = \prod_{n(\neq m)}^{M} \frac{u_m - u_n + i}{u_m - u_n - i}. \tag{55}$$

Generally, for a spin-$s$ site the product on the left-hand side of the Bethe equations features shifts $\pm i s$ [63, 64]. Thus the first factor in (55),

$$\frac{u_m - \theta_j + i/2}{u_m - \theta_j - 3i/2} = \frac{u_m - (\theta_j + i/2) + i}{u_m - (\theta_j + i/2) - i}, \tag{56}$$

corresponds to a site with inhomogeneity $\theta_j + i/2$ and spin 1. That is, the solutions to (55) with $M \leqslant L/2$ describe the eigenvectors *outside* the invariant subspace (54). (Since the transfer matrix is not Hermitian, their overlap with (54) may or may not be zero, and indeed both happen in practice.)

- The remaining eigenstates, i.e. those inside the invariant subspace (54), can be constructed using the special root

$$u_0 \equiv \theta_j + i/2, \tag{57}$$

to get from $|0\rangle$ into the invariant subspace. Indeed, we have (see Appendix C)

$$|0'\rangle \equiv \bar{B}(u_0)|0\rangle \propto |j\rangle\!\rangle - |j+1\rangle\!\rangle = (\sigma_j^- - \sigma_{j+1}^-)|\uparrow \cdots \uparrow\rangle \in V_{\text{inv}}. \tag{58}$$

By invariance of $V_{\text{inv}}$, the vector $|0'\rangle$ has Yangian highest weight, i.e. is an eigenvector of $\bar{A}, \bar{D}$ and killed by $\bar{C}$. It is thus a suitable vacuum for the algebraic Bethe Ansatz inside $V_{\text{inv}}$. Then, all the vectors in the invariant subspace can be constructed as

$$\bar{B}(u_1) \cdots \bar{B}(u_M)|0'\rangle = \bar{B}(u_0)\bar{B}(u_1) \cdots \bar{B}(u_M)|0\rangle \in V_{\text{inv}}, \tag{59}$$

where the $u_1, \ldots, u_M$ solve the 'reduced' Bethe equations with $M \leqslant (L-2)/2$

$$\prod_{i(\neq j, j+1)}^{L} \frac{u_m - \theta_i + i/2}{u_m - \theta_i - i/2} = \prod_{n(\neq m)}^{M} \frac{u_m - u_n + i}{u_m - u_n - i}, \tag{60}$$

for a spin chain with $L-2$ sites. Since the invariant subspace is still generated from a suitable pseudovacuum ('cyclic vector') the proof of completeness of [9] applies.

Let us elaborate on the special Bethe root (57). One way to understand its appearance is from the algebraic Bethe Ansatz, see Appendix B.2. For our purposes another proof is more convenient, using the QQ-relation (28). Due to the special values of inhomogeneities, it admits a class of solutions where $\bar{Q}$ and $\tilde{\bar{Q}}$ both have $u_0$ as a root:

$$\bar{Q}(u) = (u - \theta_j - i/2)\bar{Q}_{\mathrm{red}}(u), \qquad \tilde{\bar{Q}}(u) = (u - \theta_j - i/2)\tilde{\bar{Q}}_{\mathrm{red}}(u), \tag{61}$$

where $\bar{Q}_{\mathrm{red}}$ and $\tilde{\bar{Q}}_{\mathrm{red}}$ are of the form (24) and (27) with $\kappa = 1$. All three terms of the QQ-relation now have a factor $(u - \theta_j)(u - \theta_j - i)$. After removing it, we are left with

$$\bar{Q}_{\mathrm{red}}^{+} \tilde{\bar{Q}}_{\mathrm{red}}^{-} - \bar{Q}_{\mathrm{red}}^{-} \tilde{\bar{Q}}_{\mathrm{red}}^{+} = \bar{Q}_{\theta, \mathrm{red}}, \tag{62}$$

where

$$\bar{Q}_{\theta, \mathrm{red}}(u) = \prod_{i(\neq j, j+1)}^{L} (u - \theta_i). \tag{63}$$

But this is just the QQ-relation of a spin chain of length $L-2$. Thus solutions consist of the fixed root $u_0 = \theta_j + i/2$ together with $u_1, \ldots, u_M$ solving Bethe equations for an effective spin chain of effective length $L-2$ and inhomogeneities $\{\theta_1, \ldots, \theta_L\} \setminus \{\theta_j, \theta_{j+1}\}$.

Notice that the transfer-matrix eigenvalue factorises for states with the special Bethe root (57): plugging (61) into (25) we find

$$\bar{\tau}(u) = (u - \theta_j + i/2)(u - \theta_j - 3i/2) \frac{\bar{Q}_{\mathrm{red}}^{++} \bar{Q}_{\theta, \mathrm{red}}^{-} + \bar{Q}_{\mathrm{red}}^{--} \bar{Q}_{\theta, \mathrm{red}}^{+}}{\bar{Q}_{\mathrm{red}}}. \tag{64}$$

The fraction is a polynomial on shell, i.e. on solutions of the Bethe equations. Thus for states inside $V_{\mathrm{inv}}$ the eigenvalue of the transfer matrix consists of a simple factor, corresponding to the two-site singlet, times a nontrivial part due to an 'effective' spin chain with $L-2$ sites.

The reason why the fixed root is not visible in the Bethe equations (55) is that it corresponds to the vanishing of the factor $u_m - \theta_j - i/2$ that we have cancelled on the left-hand side in going from (21) to (55). In the QQ-relation, however, this root is not missed and can be treated on equal footing with the other Bethe roots. This discrepancy between the QQ-relation and solutions of the Bethe equations is explained by the fact that the usual derivation of the Bethe equations from the QQ-relation fails in this case, because $\bar{Q}$ and $\tilde{\bar{Q}}$ have a common root. For more details see Appendix B, where we also illustrate a subtlety in the proof of the construction (19) of the eigenstates for the case with the fixed root $u_0$ — namely, the 'unwanted' terms in the standard proof of the algebraic Bethe Ansatz do not cancel but rather vanish individually, providing another explanation why the explicit root is absent in the Bethe equations (see also section 6 in [65] for related discussions).

Finally, for later use we record that the special root (57) admits the symmetric expression

$$u_0 = \frac{\theta_j + \theta_{j+1}}{2}. \tag{65}$$

In Appendix D we illustrate various features of fusion into a singlet in the examples of spin chains of length $L = 2, 4$.

### 2.4.3 Bethe Ansatz for fusion into triplet

Now we consider the case of fusion into a spin-1 (triplet) representation, with

$$\theta_{j+1} = \theta_j - i. \tag{66}$$

Here completeness fails and the situation is trickier. The main features are as follows.

- The triplet in combination with the other $L-2$ sites form a Yangian-invariant subspace of $\mathcal{H}$ of dimension $3 \times 2^{L-2}$,

$$V_{\text{inv}} = \Pi^+_{j,j+1}(\mathcal{H}) \cong V_2^{\otimes(j-1)} \otimes V_3 \otimes V_2^{\otimes(L-j-1)}. \tag{67}$$

- Since the spectrum of the transfer matrix is symmetric in the inhomogeneities (see Section 2.4.1), the eigenvalues are the same as for the case with fusion into a singlet.[10]

- The eigenstates inside $V_{\text{inv}}$, which contain the reference state $|0\rangle = |\uparrow \cdots \uparrow\rangle$, are given by Bethe Ansatz as usual. The Bethe equations (21) become

$$\frac{u_m - \theta_j + 3i/2}{u_m - \theta_j - i/2} \prod_{i(\neq j,j+1)}^{L} \frac{u_m - \theta_i + i/2}{u_m - \theta_i - i/2} = \prod_{n(\neq m)}^{M} \frac{u_m - u_n + i}{u_m - u_n - i}, \tag{68}$$

and we consider all $M \leqslant L/2$, yielding states via the algebraic Bethe Ansatz (19). The prefactor

$$\frac{u_m - \theta_j + 3i/2}{u_m - \theta_j - i/2} = \frac{u_m - (\theta_j - i/2) + i}{u_m - (\theta_j - i/2) - i}, \tag{69}$$

corresponds to a spin $s = 1$ site like before, but with 'effective inhomogeneity' shifted in the other way. For both types of fusion, the effective inhomogeneity is the average $(\theta_j + \theta_{j+1})/2$ of the original inhomogeneities (cf. Appendix A).

- Crucially, the eigenstates of the transfer matrix outside the invariant space cannot be generated from $|0\rangle \in V_{\text{inv}}$ by applying $B$-operators. Thus the algebraic Bethe Ansatz (19) misses $2^{L-2}$ eigenstates: unlike for fusion into a singlet, it is *not* complete. Although the special Bethe root (65) still appears among the solutions of the QQ-relations, it is not useful this time: the $B$-operator at this value now kills the vacuum (Appendix C),

$$\bar{B}\left(\frac{\theta_j + \theta_{j+1}}{2}\right)|0\rangle = 0. \tag{70}$$

There does not seem to be a simple way to build the eigenstates outside $V_{\text{inv}}$.[11] Luckily, the applications of fusion that we will need in in Section 3 only involve vectors in the invariant subspace, so we will never need to worry about the quotient.

In appendix D.1 we illustrate some features of the fusion into a triplet on the simple example of a length $L = 2$ spin chain.

---

[10]To be precise, the sets of all eigenvalues of $\bar{t}(u; \kappa; \ldots, \theta_j, \theta_j - i, \ldots)$ and $\bar{t}(u; \kappa; \ldots, \theta_j - i, \theta_j, \ldots)$ coincide. Of course this is not true for their restrictions to the corresponding invariant subspaces.

[11]One way to describe them is to form the quotient space $\mathcal{H}/V_{\text{inv}}$. As a vector space it is isomorphic to $\Pi^-_{j,j+1}(\mathcal{H})$. Unlike the latter, the quotient *is* a well-defined Yangian representation by invariance of $V_{\text{inv}} = \Pi^+_{j,j+1}(\mathcal{H})$. This quotient corresponds to a spin chain with a spin-0 site and $L-2$ spin-1/2 sites, just like the invariant space was for fusion into a singlet. Inside the quotient one can build all eigenstates via the algebraic Bethe Ansatz as usual. These states are in one-to-one correspondence with the remaining eigenvectors of our original spin chain in $\mathcal{H}$, yet actually reconstructing them inside $\mathcal{H}$ is tricky in practice. For each state from the quotient one then needs to find an appropriate correction by a vector in $V_{\text{inv}}$. It seems rather nontrivial to do it in a systematic way.

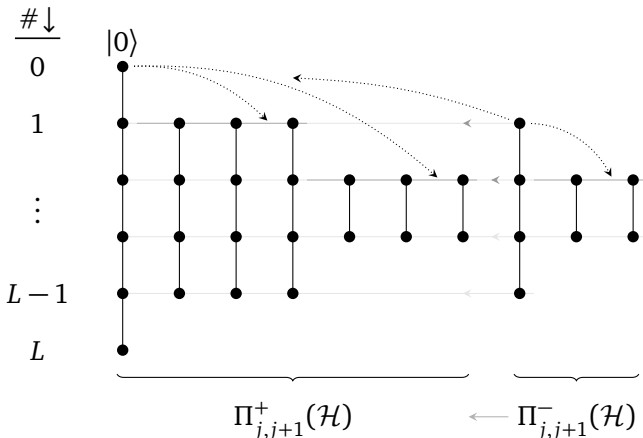

Figure 4: Structure of $\mathcal{H}$ (cf. Figure 2) with $\theta_{j+1} = \theta_j - i$ and other $\theta_i$ generic. The Yangian action preserves $V_{\text{inv}} = \Pi^+_{j,j+1}(\mathcal{H})$, in which the algebraic Bethe Ansatz works as usual, but may send vectors in $\Pi^-_{j,j+1}(\mathcal{H})$ anywhere in $\mathcal{H}$.

### 2.4.4 Repeated fusion

So far we have only looked at fusion of two sites, but fusion can happen at multiple sites. This in particular provides a way to construct an integrable Heisenberg spin chain with (possibly varying) higher-spin sites using many spin-1/2 representations and fusion [56, 60, 66]. More generally, fusion may lead to rather intricate combinations of invariant subspaces. Since we will only be interested later in some simple cases, and a general discussion would lead us too far from our goal, we merely illustrate the various possibilities for fusion when $\theta_i = \theta_j + i$ for only two pairs $(i, j)$. Up to a similarity transformation using (47), we may assume that these pairs are $(1, 2)$ and either $(2, 3)$ or $(3, 4)$—provided the chain has length $L \geqslant 4$.

In all scenarios, by the discussion in Section 2.4.1 there are at least two Yangian-invariant subspaces: $V^{(1)}_{\text{inv}} = \Pi^\pm_{12}(\mathcal{H})$ and either $V^{(2)}_{\text{inv}} = \Pi^{\pm'}_{23}(\mathcal{H})$ or $V^{(2)}_{\text{inv}} = \Pi^{\pm'}_{34}(\mathcal{H})$ (independent signs $\pm$ and $\pm'$), corresponding to fusion for either pair of sites. This time, however, these invariant spaces are not necessarily irreducible, since one could fuse both pairs at the same time. The intersection $V^{(1)}_{\text{inv}} \cap V^{(2)}_{\text{inv}}$ is either trivial or a Yangian-irreducible subspace. Here are the various possibilities:

- **Independent fusion.** When $\theta_2 = \theta_1 \mp i$ and $\theta_4 = \theta_3 \mp' i$ (independent signs), we get fusion separately at sites $1, 2$ and at sites $3, 4$. Here $V^{(i)}_{\text{inv}}$ are reducible but indecomposable, since both contain the nontrivial subspace

$$\Pi^\pm_{12} \Pi^{\pm'}_{34}(\mathcal{H}) = \Pi^\pm_{12}(\mathcal{H}) \cap \Pi^{\pm'}_{34}(\mathcal{H}). \tag{71}$$

This Yangian-invariant subspace is irreducible and can be viewed as a spin chain of length $L - 2$ containing two sites with spin 1 or 0, depending on the signs.

In particular, $\theta_2 = \theta_1 + i$, $\theta_4 = \theta_3 + i$ leaves us with (two spin-0 sites and) $L - 4$ spin-1/2 sites. This is essentially what will happen for the fermionic spin-Calogero–Sutherland model in the following sections. (The case $\theta_2 = \theta_1 - i$, $\theta_4 = \theta_3 - i$ would instead appear if one were to consider the bosonic spin-Calogero–Sutherland model, cf. [23].)

- **Three-site antisymmetric fusion.** The case $\theta_3 = \theta_2 + i = \theta_1 + 2i$ corresponds to singlet fusion at sites $1, 2$ as well as at sites $2, 3$. Both $V^{(i)}_{\text{inv}}$ are irreducible as their intersection

$$\Pi^-_{12}(\mathcal{H}) \cap \Pi^-_{23}(\mathcal{H}) = \{0\}, \tag{72}$$

is trivial because there is no completely antisymmetric tensor with three indices taking only two values.[12] Such a situation will occur in Section 4.3.2.

- **Three-site symmetric fusion.** The case $\theta_3 = \theta_2 - i = \theta_1 - 2i$ corresponds to triplet fusion at sites $1, 2$ as well as $2, 3$. Both $V_{\text{inv}}^{(i)}$ are reducible and indecomposable, since the intersection

$$\Pi_{12}^+(\mathcal{H}) \cap \Pi_{23}^+(\mathcal{H}) \cong V_4 \otimes V_2^{\otimes(L-3)}, \tag{73}$$

  is a non-trivial irreducible Yangian submodule. It is the space of a spin chain of length $L-2$ with one spin-3/2 site. (This scenario would also show up for the bosonic spin-Calogero–Sutherland model.)

- **Three-site mixed fusion.** For $\theta_3 = \theta_2 \mp i = \theta_1$ both $V_{\text{inv}}^{(i)}$ are irreducible, as

$$\Pi_{12}^\pm(\mathcal{H}) \cap \Pi_{23}^\mp(\mathcal{H}) = \{0\}, \tag{74}$$

  is again trivial, because there is no 3-tensor that is symmetric in its first (last) two indices and antisymmetric in the last (first) two. Since $\dim\big(V_{\text{inv}}^{(1)}\big) + \dim\big(V_{\text{inv}}^{(2)}\big) = \dim(\mathcal{H})$, now $\mathcal{H}$ is actually completely reducible: $\mathcal{H} = \Pi_{12}^\pm(\mathcal{H}) \oplus \Pi_{23}^\mp(\mathcal{H})$ as Yangian modules.

One can continue in this way to get more and more complicated constellations of invariant subspaces. As the scenarios with three-site fusion illustrate, one can use this to construct a spin chain with sites of varying spins, see [56, 60, 66] for more details and examples. For us only the case of independent fusion will be relevant in what follows. As the preceding illustrates, for any number of nonoverlapping pairs of neighbouring sites one can essentially omit the fused singlets to get a spin chain of length $L - 2n$ for some $n$. As we will see, the fermionic spin-Calogero–Sutherland model contains *infinitely* many of these spin chains.

Equipped with these preliminaries we are ready to move on to our main subject.

## 3 Fermionic Spin-Calogero–Sutherland model

The (trigonometric, quantum) spin-Calogero–Sutherland model is a quantum many-body system describing particles that carry a spin and move around on a circle while interacting in pairs. It is a (quantum) integrable model with extraordinary properties, including extremely simple eigenvalues that are highly degenerate because of a Yangian *symmetry*. This should be contrasted with the Heisenberg spin, whose Yangian does not commute with the spin-chain Hamiltonian and instead allows one to move between different eigenspaces, as in the algebraic Bethe Ansatz. The algebraic origin of the spin-Calogero–Sutherland model and its properties lies in a family of commuting differential-difference operators known as the Dunkl operators.

**Conventions.** We will clean up our notation a little from now on. Let us summarise the changes for easy reference. To remove the factors of i that were floating around in Section 2 we reparametrise the spectral parameter as $u = i x$ and henceforth use the (slightly differently normalised) $R$-matrix

$$R(x) \equiv 1 + x^{-1} P = 1 + i u^{-1} P = u^{-1} \bar{R}(u). \tag{75}$$

---

[12]This is a peculiarity of the low-rank chain we are considering. If we were studying an $\mathfrak{sl}_r$ spin chain for $r > 2$, then the intersection (72) would be a non-trivial Yangian-irreducible subspace. It would be the space of a spin chain of length $L-2$ in which the first site carries the (third fundamental) $\mathfrak{sl}_r$-irrep whose highest-weight vector is a completely antisymmetric 3-tensor. For $r = 3$ it is the trivial representation of $\mathfrak{sl}_3$.

We reparametrise the inhomogeneities as $\theta_j = -\mathrm{i}\,\delta_j$; in practice, $\delta_i$ will either be a Dunkl operator or its (real) eigenvalue. The monodromy matrix thus takes the form of a product of $R_{0i}(x + \delta_i - 1/2) = \bar{R}_{0i}(u - \theta_i - \mathrm{i}/2)/(u - \theta_i - \mathrm{i}/2)$, cf. (94) below. Note that the fusion condition $\theta_{j+1} - \theta_j = \mp\mathrm{i}$ from Section 2.4 now reads $\delta_{j+1} - \delta_j = \pm 1$ (triplet/singlet). The Bethe roots become $u_m = \mathrm{i}\,x_m$. We reserve $L$ for the lengths of the 'effective spin chains' to appear in Section 3.3, and start with $N$ particles for our quantum many-body system.

## 3.1 Dunkl operators and nonsymmetric Jack polynomials

Let $\mathbb{C}[z_1^{\pm}, \ldots, z_N^{\pm}]$ be the space of complex Laurent polynomials in $N$ variables, which will be the coordinates of the particles on the circle in multiplicative notation, $z_j = \mathrm{e}^{\mathrm{i}x_j}$. We denote the operators of coordinate permutation $z_i \leftrightarrow z_j$ by $s_{ij}$ for $i \neq j$ in $\{1, \ldots, N\}$. For $\beta \in \mathbb{C} \setminus \{0\}$, the Dunkl operators are

$$d_i = \frac{1}{\beta}\,z_i\,\partial_{z_i} - \sum_{j=1}^{i-1} \frac{z_i}{z_{ji}}\,(1 - s_{ij}) + \sum_{j=i+1}^{N} \frac{z_j}{z_{ij}}\,(1 - s_{ij}) + \frac{N + 1 - 2\,i}{2}\,, \tag{76}$$

where we use the abbreviation $z_{ij} \equiv z_i - z_j$. Their key properties are the (degenerate affine Hecke algebra) relations

$$d_i\,d_j = d_j\,d_i\,, \qquad d_i\,s_{i,i+1} = s_{i,i+1}\,d_{i+1} + 1\,, \qquad d_i\,s_{jk} = s_{jk}\,d_i\,, \quad \text{for} \quad i \neq j, k\,. \tag{77}$$

Dunkl's operators have a simple joint spectrum, with simultaneous eigenfunctions that are called nonsymmetric Jack polynomials $E_{\boldsymbol{\mu}}(\boldsymbol{z}) = E_{\boldsymbol{\mu}}^{(\alpha)}(\boldsymbol{z})$, with Jack parameter $\alpha = 1/\beta$ that we suppress. These polynomials are indexed by 'compositions' $\boldsymbol{\mu} = (\mu_1, \ldots, \mu_N) \in \mathbb{Z}^N$, and defined by the conditions

$$E_{\boldsymbol{\mu}}(\boldsymbol{z}) = z_1^{\mu_1} \cdots z_N^{\mu_N} + \text{lower}\,, \qquad \begin{aligned} d_i\,E_{\boldsymbol{\mu}}(\boldsymbol{z}) &= \delta_i(\boldsymbol{\mu})\,E_{\boldsymbol{\mu}}(\boldsymbol{z})\,, \\ \delta_i(\boldsymbol{\mu}) &\equiv \frac{\mu_i}{\beta} + \frac{1}{2}\big(N + 1 - 2\,\sigma^{\boldsymbol{\mu}}(i)\big)\,. \end{aligned} \tag{78}$$

Here 'lower' indicates monomials that are lower in the dominance order on compositions (see e.g. Section 2.1 of [41]). The eigenvalues $\delta_j(\boldsymbol{\mu})$ of the Dunkl operators contain the integers

$$\sigma^{\boldsymbol{\mu}}(i) \equiv \#\big\{1 \leqslant j \leqslant N \,\big|\, \mu_j > \mu_i\big\} + \#\big\{1 \leqslant j \leqslant i \,\big|\, \mu_j = \mu_i\big\}\,, \tag{79}$$

Note that $\sigma^{\boldsymbol{\mu}}(i) = i$ if $\mu_1 \geqslant \cdots \geqslant \mu_N$. We will further need the property that, for all $\boldsymbol{\mu} \in \mathbb{Z}^N$,

$$\delta_i(\boldsymbol{\mu}) = \delta_{i+1}(\boldsymbol{\mu}) + 1\,, \quad \text{if} \quad \mu_i = \mu_{i+1}\,. \tag{80}$$

The Dunkl operators give rise to the spin-Calogero–Sutherland model through intermediate operators defined as symmetric combinations of the $d_j$. These in particular include the 'gauge-transformed (total) momentum operator'

$$P' \equiv \beta \sum_{i=1}^{N} d_i = \sum_{i=1}^{N} z_i\,\partial_{z_i}\,, \tag{81}$$

and the 'gauge-transformed Hamiltonian'

$$\begin{aligned} H' &\equiv \frac{\beta^2}{2}\left(\sum_{i=1}^{N} d_i^2 - E^0\right) \\ &= \frac{1}{2}\sum_{i=1}^{N}\big(z_i\,\partial_{z_i}\big)^2 + \frac{\beta}{2}\sum_{i<j} \frac{z_i + z_j}{z_i - z_j}\big(z_i\,\partial_{z_i} - z_j\,\partial_{z_j}\big) + \beta \sum_{i<j} \frac{z_i\,z_j}{z_{ij}\,z_{ji}}\,(1 - s_{ij})\,, \end{aligned} \tag{82}$$

where we defined the constant

$$E^0 \equiv \frac{1}{4} \sum_{i=1}^{N} (N - 2i + 1)^2 = \frac{1}{12} N (N^2 - 1) .$$ (83)

The reason for the adjective 'gauge-transformed' is that they are related to the true (continuum) momentum operator and Hamiltonian by conjugation:[13]

$$P' = \Phi_0^{-1} \left( \sum_{i=1}^{N} z_i \, \partial_{z_i} \right) \Phi_0 , \qquad \Phi_0(z) \equiv \prod_{i \neq j}^{N} (1 - z_i/z_j)^{\beta/2} ,$$

$$H' + \frac{\beta^2}{2} E^0 = \Phi_0^{-1} \left( \frac{1}{2} \sum_{i=1}^{N} (z_i \, \partial_{z_i})^2 + \sum_{i<j} \frac{z_i z_j}{z_{ij} z_{ji}} \beta (\beta - s_{ij}) \right) \Phi_0 .$$ (84)

The eigenvalues of these operators only depend on the partition $\lambda$ obtained by sorting the parts of $\mu$ into (weakly) decreasing order. From the definition (82) of the gauge-transformed operators, it is clear that these eigenvalues can only be of the form

$$P'(\mu) = \beta \sum_{i=1}^{N} \delta_i(\mu) = \sum_{i=1}^{N} \lambda_i ,$$

$$E'(\mu) = \frac{\beta^2}{2} \left( \sum_{i=1}^{N} \delta_i(\mu)^2 - E^0 \right) = \frac{1}{2} \sum_{i=1}^{N} \lambda_i^2 + \frac{\beta}{2} \sum_{i=1}^{N} (N - 2i + 1) \lambda_i .$$ (85)

The integers $\mu_i$ can be interpreted as 'quantum numbers' for the (quasi)momenta of the quasiparticles.

So far we have worked at the nonsymmetric level, corresponding to *dis*tinguishable particles. The spectrum of this model is highly degenerate: the eigenvalues (85) only depend on the partition $\lambda$. By prescribing the symmetry of the eigenvectors one obtains *in*distinguishable particles. For example, for spinless bosons (or fermions) the wave functions are completely (anti)symmetric, and on the subspaces of totally (anti)symmetric Laurent polynomials one recovers the scalar bosonic (fermionic) trigonometric Calogero–Sutherland model,

$$P = \Phi_0 P' \Phi_0^{-1} = -i \sum_{i=1}^{N} \partial_{x_i} , \qquad\qquad z_i = e^{i x_i} ,$$

$$H = \Phi_0 H' \Phi_0^{-1} = -\frac{1}{2} \sum_{i=1}^{N} \partial_{x_i}^2 + \beta (\beta \mp 1) \sum_{i<j} \frac{1}{4 \sin^2[(x_i - x_j)/2]} , \qquad s_{ij} = \pm 1 ,$$ (86)

where we passed to additive coordinates. The eigenfunctions can be obtained from the nonsymmetric theory too. We return to the gauge-transformed setting, where we can work with wave functions that are Laurent polynomials. Up to normalisation, the total (anti)symmetrisation of $E_\mu(z)$ only depends on the partition $\lambda$ corresponding to $\mu$, yielding single wave function with momentum and energy (85). For bosons the symmetrisation gives the (symmetric) Jack polynomial $P_\lambda(z)$ with parameter $\alpha = 1/\beta$. For fermions the result is only nonzero if all parts of $\lambda$ are different, which is because the non-symmetric Jack polynomials obey

$$s_{i,i+1} E_\lambda = E_\lambda , \quad \text{if} \quad \lambda_i = \lambda_{i+1} .$$ (87)

If $\lambda$ is a *strict* partition, i.e. if $\lambda_1 > \cdots > \lambda_N$, then it is of the form $\lambda = \nu + \delta_N$ for some (not necessarily strict) partition $\nu$, where $\delta_N \equiv (N-1, \ldots, 1, 0)$ is the staircase partition.

---

[13]We avoid the adjective 'effective' that is often used instead of 'gauge transformed' to prevent any confusion with our (unrelated) term 'effective spin chain' to appear in Section 3.3.

For strict partitions the result of antisymmetrisation is a Vandermonde polynomial times a Jack polynomial with shifted parameter,

$$\text{Vand}(z_1, \ldots, z_N) \, P'_{\nu}(z), \qquad \text{Vand}(z_1, \ldots, z_N) \equiv \prod_{1 \leqslant i < j \leqslant N} (z_i - z_j), \quad P'_{\nu} \equiv P_{\nu}\big|_{\beta \mapsto \beta+1}. \tag{88}$$

In the spinless case, then, each energy in (85) occurs in the bosonic case, while in the fermionic case only strict partitions are allowed. See e.g. §2.2 in [41] and references therein for more. We will instead be interested in a generalisation with fermions that each carry a spin as well as a coordinate.

## 3.2 Hamiltonian and monodromy matrix

The Hilbert space for $N$ spin-1/2 fermions moving on a circle is

$$\mathcal{F} = \left\{ |\Psi\rangle \in \left(\mathbb{C}^2\right)^{\otimes N} \otimes \mathbb{C}\!\left[z_1^{\pm}, \ldots, z_N^{\pm}\right] \; \middle| \; P_{ij} s_{ij} |\Psi\rangle = -|\Psi\rangle \right\}. \tag{89}$$

It consists of the vectors that are completely antisymmetric in spin and coordinates, and coincides with the image of the projector

$$\Pi_-^{\text{tot}} = \frac{1}{N!} \sum_{\sigma \in S_N} \text{sgn}(\sigma) \, P_\sigma \, s_\sigma, \qquad \left(\Pi_-^{\text{tot}}\right)^2 = \Pi_-^{\text{tot}}. \tag{90}$$

On the fermionic space, the gauge-transformed Hamiltonian (82) takes the form

$$\widetilde{H}' = \frac{1}{2} \sum_{i=1}^N \left(z_i \, \partial_{z_i}\right)^2 + \frac{\beta}{2} \sum_{i<j} \frac{z_i + z_j}{z_i - z_j} \left(z_i \, \partial_{z_i} - z_j \, \partial_{z_j}\right) + \beta \sum_{i<j} \frac{z_i z_j}{z_{ij} z_{ji}} \left(1 + P_{ij}\right). \tag{91}$$

It is related by conjugation as in (86) to the fermionic spin-Calogero–Sutherland Hamiltonian (1). The momentum operator (81), on the other hand, is the same as in the spinless case. Let us emphasise that this operator only acts on the coordinates $z_j$ of the particles, not on their spins. The cyclic translation $P_{12} \cdots P_{N-1,N}$ of the spins does not even act on the fermionic space, so the spin-chain notion of (crystal) momentum is irrelevant for the spin-Calogero–Sutherland model.

The spectrum of (1) is given by (85) with the restriction that $\boldsymbol{\lambda}$ be a partition with multiplicities $\leqslant 2$. We denote the set of these allowed partitions by[14]

$$\mathcal{P} = \{\boldsymbol{\lambda} \in \mathbb{Z}^N \mid \lambda_1 \geqslant \cdots \geqslant \lambda_N, \; \lambda_i > \lambda_{i+2}\}. \tag{92}$$

Indeed, by the property (87) of nonsymmetric Jack polynomials, repetitions in $\boldsymbol{\lambda}$ require antisymmetry in the corresponding spins because of the fermionic condition (89). For our case of spin 1/2 (i.e. $\mathfrak{sl}_2$), this means that the multiplicities are at most 2. (For spinless fermions the multiplicities are at most 1.)

The fermionic space comes equipped with (an action of the Yangian of $\mathfrak{gl}_2$ given by) the monodromy matrix [38]

$$\begin{aligned} T_0(x) &= R_{01}\!\left(x + d_1 - \tfrac{1}{2}\right) \cdots R_{0N}\!\left(x + d_N - \tfrac{1}{2}\right) \\ &= \left(1 + \frac{P_{01}}{x + d_1 - \tfrac{1}{2}}\right) \cdots \left(1 + \frac{P_{0N}}{x + d_N - \tfrac{1}{2}}\right). \end{aligned} \tag{93}$$

---

[14]Such partitions are called '3-regular' in representation theory – not to be confused with the different but related meaning of that term in combinatorics, cf. https://mathoverflow.net/q/438228.

Here we use the *R*-matrix (75), and Dunkl operators play the role of inhomogeneities: in terms of the conventions of Section 2 one has

$$T_0(x) = \left.\frac{\overline{T}_0(\mathrm{i} x)}{\overline{Q}_\theta\left(\mathrm{i} x - \frac{\mathrm{i}}{2}\right)}\right|_{\theta=-\mathrm{i}\boldsymbol{d}}. \tag{94}$$

In [38] the term 'quantised inhomogeneities' was used to emphasise that the inhomogeneities are now nontrivial operators (on polynomials). The relations (77) guarantee it preserves the fermionic space (see e.g. §C of [41]) and obeys the *RTT* relations. The proper representation-theoretic meaning of (93) stems from affine Schur–Weyl duality [21].

There are several ways to make sense of the Dunkl operators in the denominators in (93): (i) expanding as a formal power series in $x^{-1}$, (ii) using the nonsymmetric Jack basis for the polynomial factor of the fermionic space to replace the Dunkl operators by their eigenvalues, (iii) removing the denominator $\prod_j(x + d_j - 1/2)$, which is central and acts in a simple way. The third point is related to the following important property of the monodromy matrix.

The spin-Calogero–Sutherland model commutes with the Yangian action given by (93). Indeed, the hierarchy of spin-Calogero–Sutherland Hamiltonians [32,40,42] are generated by the quantum determinant [38,40]

$$\begin{aligned}
\Delta(x) = \mathrm{qdet}_0 T_0(x) &= \prod_{i=1}^{N} \frac{x + d_i + \frac{1}{2}}{x + d_i - \frac{1}{2}} \\
&= 1 + N x^{-1} + \left(\frac{N^2}{2} - \frac{P'}{\beta}\right) x^{-2} + \left(\frac{N^3}{4} - \frac{N P'}{\beta} + \frac{2H'}{\beta^2}\right) x^{-3} + O\left(x^{-4}\right),
\end{aligned} \tag{95}$$

and the quantum determinant of the Yangian generates its centre.

Let us finally mention that for $\beta > 0$ one can define a scalar product on $\mathbb{C}[z_1^\pm, \ldots, z_N^\pm]$ for which the Dunkl operators are Hermitian, see Proposition 3.8 in [67] and §2 of [68]. The natural extension of this scalar product to the fermionic space $\mathcal{F}$ defined in (89) is such that the Yangian algebra is stable under Hermitian conjugation.[15] This implies in particular that the Yangian representation on the fermionic space is completely reducible. The decomposition of $\mathcal{F}$ into irreducible components is our next topic.

## 3.3 Effective spin chains

In [23] it was shown that the Hilbert space of the fermionic spin-Calogero–Sutherland model decomposes into a sum of irreducible representations of the Yangian:

$$\mathcal{F} = \bigoplus_{\lambda \in \mathcal{P}} \mathcal{F}_\lambda. \tag{96}$$

The summands are also eigenspaces for the spin-Calogero–Sutherland model. The momentum operator (81) and gauge-transformed Hamiltonian (91) still have eigenvalues (85).

The eigenspace $\mathcal{F}_\lambda$ is the image by the projector (90) of the subspace which, in the polynomial factor, is spanned by all nonsymmetric Jack polynomials $E_\mu(\boldsymbol{z})$ with composition $\mu \in \mathbb{Z}^N$ differing from the partition $\lambda$ by reordering:

$$\mathcal{F}_\lambda = \Pi_{\mathrm{tot}}^- \left( \bigoplus_{\mu \in S_N \cdot \lambda} E_\mu(\boldsymbol{z}) \otimes (\mathbb{C}^2)^{\otimes N} \right) \subset \mathcal{F}. \tag{97}$$

Following [23], this subspace can be equivalently viewed as an 'effective spin chain' of some length $L_\lambda \leqslant N$ with particular (scalar) inhomogeneities. This goes as follows.

---

[15]More precisely, if we expand the A-, ..., D-operators in (93) as $A(x) = 1 + \sum_{n=1}^{+\infty} A_n x^{-n}$, $B(x) = \sum_{n=1}^{+\infty} B_n x^{-n}$, etc., then the coefficients obey $A_n^\dagger = A_n$, $D_n^\dagger = D_n$ and $B_n^\dagger = C_n$ [23].

Let us set up some notation. For each allowed partition $\lambda \in \mathcal{P}$, we define sets $I_\lambda$ and $J_\lambda$ that enumerate its unique and repeated parts, respectively:

$$I_\lambda \equiv \left\{ 1 \leqslant i \leqslant N \mid \lambda_{i-1} > \lambda_i > \lambda_{i+1} \right\}, \qquad J_\lambda \equiv \left\{ 1 \leqslant j < N \mid \lambda_j = \lambda_{j+1} \right\}, \qquad (98)$$

with the convention that $\lambda_0 \equiv +\infty$ and $\lambda_{N+1} \equiv -\infty$. If $\lambda = (7,6,6,2,2,-5,-6,-6,-8)$, for instance, then $I_\lambda = \{1,6,9\}$ and $J_\lambda = \{2,4,7\}$. The set $J_\lambda$ is called a *motif*. $I_\lambda$ will label the sites of the effective chain, while $J_\lambda$ will record pairs of sites of the original chain that are fused into singlets. In particular, the effective length will be

$$L_\lambda \equiv \#I_\lambda = N - 2\,\#J_\lambda. \qquad (99)$$

We start with the Yangian highest-weight vector in $\mathcal{F}_\lambda$. It contains $M_\lambda \equiv \#J_\lambda$ magnons, cf. (87), and can be written as

$$|0_\lambda\rangle \propto \Pi_{\text{tot}}^- \left( E_{\bar{\lambda}}(z) \, |1, \ldots, M_\lambda\rangle\!\rangle \right), \qquad (100)$$

where we allowed for a normalising constant, and $\bar{\lambda}$ is any rearrangement of $\lambda$ such that the result of antisymmetrising is nonzero.[16] Like for any $M_\lambda$-magnon fermionic vector, (100) can be recast in the form (see §2.3.1 of [41])

$$|0_\lambda\rangle = \sum_{j_1 < \cdots < j_{M_\lambda}} (-1)^{\sum_m (j_m - m)} f_\lambda\big(z_{j_1}, \ldots, z_{j_{M_\lambda}}; \text{other } z\text{'s}\big) \, |j_1, \ldots, j_{M_\lambda}\rangle\!\rangle, \qquad (101a)$$

where, because of (100),

$$f_\lambda(z_1, \ldots, z_{M_\lambda}; z_{M_\lambda+1}, \ldots, z_N) = \langle\!\langle 1 \cdots M_\lambda | 0_\lambda\rangle \propto \sum_{\sigma \in S_{M_\lambda} \times S_{N-M_\lambda}} \text{sgn}(\sigma) \, E_{\bar{\lambda}}(z_{\sigma(1)}, \ldots, z_{\sigma(N)}), \qquad (101b)$$

is a partially antisymmetrised nonsymmetric Jack polynomial, again up to a constant that depends on the choice of $\bar{\lambda}$. Note that it will be divisible by the partial Vandermonde factor $\text{Vand}(z_1, \ldots, z_{M_\lambda}) \, \text{Vand}(z_{M_\lambda+1}, \ldots, z_N)$. The expression (101) was obtained by two of us [41], and should be contrasted with equation (5.31) in [23]: our expression is given in the coordinate basis of $V_2^{\otimes N}$ and has nontrivial polynomial coefficients, whilst Takemura–Uglov use the nonsymmetric Jack basis of $\mathbb{C}[z_1^\pm, \ldots, z_N^\pm]$ and has a nontrivial spin coefficient.

Here are some examples of the highest-weight vectors in the fermionic space. If $\lambda$ is a strict partition, i.e. $\lambda_i > \lambda_{i+1}$ for all $i$, so that $J_\lambda = \varnothing$ and $M_\lambda = 0$, then the highest-weight vector acquires the simple form $|0_\lambda\rangle = \text{Vand}(z_1, \ldots, z_N) \, P'_\nu(z) \, |\uparrow \cdots \uparrow\rangle$ because of (88). For this example the effective spin chain will have length $L_\lambda = N$ and can be viewed as $2^N$ copies of the spinless fermionic Calogero–Sutherland model, with degeneracies due to the Yangian symmetry. Another class of easy examples occurs when $I_\lambda = \varnothing$, i.e. when $N$ is even and $\lambda_i = \lambda_{i+1}$ for all odd $i$. The corresponding effective spin chain has length $L_\lambda = 0$ (after fusion), i.e. a one-dimensional Hilbert space. For instance, $\lambda = (N/2 - 1, N/2 - 1, \ldots, 1, 1, 0, 0)$ gives a vector of the form (101) at the equator $M_\lambda = N/2$ with $f_\lambda = \text{Vand}(z_1, \ldots, z_{N/2}) \, \text{Vand}(z_{N/2+1}, \ldots, z_N)$ as can be seen by counting the degree.

From the highest-weight vector $|0_\lambda\rangle$ one obtains the rest of the fermionic eigenspace $\mathcal{F}_\lambda$ by acting with the monodromy matrix (93). Takemura and Uglov [23] gave an explicit description of the Yangian structure of $\mathcal{F}_\lambda$.[17] Namely, first consider a chain with $N$ spin-1/2

---

[16] In particular, this requires $\{\bar{\lambda}_1, \ldots, \bar{\lambda}_{M_\lambda}\} = J_\lambda$. One choice is to order $\bar{\lambda}$ such that $\bar{\lambda}_1 > \cdots > \bar{\lambda}_{M_\lambda}$ and $\bar{\lambda}_{M_\lambda+1} > \cdots > \bar{\lambda}_N$, as in [41]. Another one instead has $\bar{\lambda}_1 < \cdots < \bar{\lambda}_{M_\lambda}$ and $\bar{\lambda}_{M_\lambda+1} < \cdots < \bar{\lambda}_N$, which is a little more efficient (cf. Section 4.4). In any case, different choices of $\bar{\lambda}$ only affect the normalisation of (100).

[17] In representation-theoretic terms, any finite-dimensional Yangian irrep is isomorphic to a tensor product of evaluation modules (see e.g. §12.1.E in [62]). Here we interpret this in physical terms as an inhomogeneous Heisenberg chain as in Section 2.

sites, with ('ambient') Hilbert space $V_2^{\otimes N}$ and inhomogeneities $\delta_1(\boldsymbol{\lambda}), \ldots, \delta_N(\boldsymbol{\lambda})$ equal to the eigenvalues (78) of the Dunkl operators, which for partitions are given by

$$\delta_i(\boldsymbol{\lambda}) = \frac{\lambda_i}{\beta} + \frac{1}{2}(N + 1 - 2i). \tag{102}$$

Thus the monodromy matrix reads

$$R_{01}(x + \delta_1(\boldsymbol{\lambda}) - 1/2) \cdots R_{0N}(x + \delta_N(\boldsymbol{\lambda}) - 1/2). \tag{103}$$

By Sections 2.4.2 and 2.4.4, singlet fusion happens whenever $\boldsymbol{\lambda}$ has repeats. The invariant subspace thus has $M_{\boldsymbol{\lambda}} = \#J_{\boldsymbol{\lambda}}$ sites with spin 0, and $L_{\boldsymbol{\lambda}} = N - 2M_{\boldsymbol{\lambda}}$ spin-1/2 sites. The highest-weight vector

$$\prod_{j \in J_{\boldsymbol{\lambda}}} (\sigma_j^- - \sigma_{j+1}^-) |0\rangle \in V_2^{\otimes N} \tag{104}$$

has singlets at sites $j, j+1$ for $j \in J_{\boldsymbol{\lambda}}$, and $\uparrow$ at all remaining sites $i \in I_{\boldsymbol{\lambda}}$. By Section 2.4.2, see (58), the vector (104) can be written in algebraic Bethe-Ansatz form by acting on $|0\rangle \in V_2^{\otimes N}$ with $B$-operators from (103) at the fixed Bethe roots $x_0^{(j)} \equiv (\delta_j(\boldsymbol{\lambda}) + \delta_{j+1}(\boldsymbol{\lambda}))/2$ for $j \in J_{\boldsymbol{\lambda}}$. Takemura–Uglov [23] constructed an isomorphism of Yangian modules between $\mathcal{F}_{\boldsymbol{\lambda}}$ and this invariant subspace. The highest-weight vector $|0_{\boldsymbol{\lambda}}\rangle \in \mathcal{F}_{\boldsymbol{\lambda}}$ from (101) corresponds to (104) under this isomorphism. Note that the remaining inhomogeneities $\theta_i$ ($i \in I_{\boldsymbol{\lambda}}$) are generic.

We can simplify the setting a little further by omitting the singlets, which brings us to our *effective spin chain*. Its Hilbert space is

$$\mathcal{H}_{\boldsymbol{\lambda}} \equiv V_2^{\otimes L_{\boldsymbol{\lambda}}}, \tag{105}$$

which serves as a 'model space' for $\mathcal{F}_{\boldsymbol{\lambda}} \subset \mathcal{F}$. The highest-weight vector $|0_{\boldsymbol{\lambda}}\rangle \in \mathcal{F}_{\boldsymbol{\lambda}}$ from (101) now simply corresponds to $|\uparrow\rangle^{\otimes L_{\boldsymbol{\lambda}}} \in \mathcal{H}_{\boldsymbol{\lambda}}$. The space (105) is isomorphic to the invariant subspace of $V_2^{\otimes N}$ as a (irreducible) representation for the Yangian. If we denote the elements of the set $I_{\boldsymbol{\lambda}}$ by $i_1 < \cdots < i_{L_{\boldsymbol{\lambda}}}$, then the Yangian acts on $\mathcal{H}_{\boldsymbol{\lambda}}$ via the monodromy matrix

$$(T_{\boldsymbol{\lambda}})_0(x) = \prod_{j \in J_{\boldsymbol{\lambda}}} \frac{x + \delta_j(\boldsymbol{\lambda}) + \frac{1}{2}}{x + \delta_j(\boldsymbol{\lambda}) - \frac{1}{2}} \times R_{01}(x + \delta_{i_1}(\boldsymbol{\lambda}) - \tfrac{1}{2}) \cdots R_{0L_{\boldsymbol{\lambda}}}(x + \delta_{i_{L_{\boldsymbol{\lambda}}}}(\boldsymbol{\lambda}) - \tfrac{1}{2}), \tag{106}$$

with prefactor coming from the $R$-matrices in (103) that have been fused into singlets (cf. Appendix A). Observe that $I_{\boldsymbol{\lambda}}$ (and $J_{\boldsymbol{\lambda}}$) was defined in (98) from the (quasi)*momentum* quantum numbers $\boldsymbol{\lambda}$, but labels the *sites* (positions) of the effective chain on $\mathcal{H}_{\boldsymbol{\lambda}}$ (and its ambient space). We stress that the spin-Calogero–Sutherland model contains infinitely many different effective spin chains, one for each allowed $\boldsymbol{\lambda} \in \mathcal{P}$.

## 4 Bethe-Ansatz analysis of the spin-Calogero–Sutherland model

We can import the standard toolkit of Heisenberg integrability from Section 2 into the world of spin-Calogero–Sutherland models from Section 3 thanks to the Takemura–Uglov isomorphism from Section 3.3. As we have seen in Section 3.2, the spin-Calogero–Sutherland Hamiltonian is invariant under the whole Yangian (93). In particular it commutes with the (twisted) transfer matrix

$$t(x; \kappa) = \text{Tr}_0 \Big[ \kappa^{\sigma_0^z} T_0(x) \Big]. \tag{107}$$

This provides a refinement of the spin-Calogero–Sutherland hierarchy: since the transfer matrix does not commute with the Yangian (just like for the Heisenberg chain in Section 2),

the Heisenberg-style Hamiltonians generated by the transfer matrix are nontrivial on $\mathcal{F}_\lambda$, lifting the degeneracies of the spin-Calogero–Sutherland model. In representation-theoretic language we pass from the quantum determinant (centre) to a Bethe subalgebra (maximal abelian subalgebra) of the Yangian that depends on the twist $\kappa$. The only spin symmetry that remains from the Yangian is $\mathfrak{sl}_2$, which is further broken down to the (Cartan sub)algebra $\mathfrak{u}_1$ generated by $S^z$ when $\kappa \neq \pm 1$.

Since the usual hierarchy (95) is proportional to the identity on each Yangian irrep in the fermionic space, any basis of $\mathcal{F}_\lambda$ provides eigenvectors of the spin-Calogero–Sutherland model. One distinguished basis is the (Yangian) Gelfand–Tsetlin basis [56, 57], which was constructed for the spin-Calogero–Sutherland model by Takemura–Uglov [23]. By diagonalising the Heisenberg-style Hamiltonians we will construct a new Bethe-Ansatz eigenbasis of the spin-Calogero–Sutherland model, which reduces to the Gelfand–Tsetlin basis in the limit of extreme twist.

## 4.1 Heisenberg-style symmetries

Let us first extract some of the refined Hamiltonians from the transfer matrix (107). The operators constructed in Section 2.3.1 are not compatible with the fermionic condition (89). Thus we proceed as in Section 2.3.2 and expand the transfer matrix as $x \to \infty$. Replacing $\theta_j \to -\mathrm{i}\, d_j$ and $u \to \mathrm{i}\, x$ in the results of Section 2.3.2, we obtain

$$
t\left(x + \frac{1}{2}; \kappa\right) = \kappa + \kappa^{-1} + \left((\kappa + \kappa^{-1})\frac{N}{2} + (\kappa - \kappa^{-1})S^z\right)x^{-1} + \left(\sum_{i<j}\kappa^{\sigma_j^z}P_{ij} - \sum_{i=1}^{N}\kappa^{\sigma_i^z}d_i\right)x^{-2}
$$

$$
+ \left(\sum_{i<j<k}\kappa^{\sigma_k^z}P_{jk}P_{ij} - \sum_{i<j}\kappa^{\sigma_j^z}P_{ij}(d_i + d_j) + \sum_{i=1}^{N}\kappa^{\sigma_i^z}d_i^2\right)x^{-3} + O\left(x^{-4}\right). \quad (108)
$$

As mentioned at the end of Section 3.2, there exists a scalar product such that all the coefficients in this expansion are Hermitian provided $\kappa$ is real. Hence, their eigenvalues must be real. When $\kappa \neq 1$, the coefficient $t_2(\kappa)$ in front of $x^{-2}$ is already a non-trivial operator acting on both the spins and the coordinates of the particles. It can be rewritten as

$$
\begin{aligned}
t_2(\kappa) &= \sum_{i<j}\kappa^{\sigma_j^z}P_{ij} - \sum_{i=1}^{N}\kappa^{\sigma_i^z}d_i \\
&= \frac{\kappa + \kappa^{-1}}{2}\left(\sum_{i<j}P_{ij} - \frac{P'}{\beta}\right) \\
&\quad + \frac{\kappa - \kappa^{-1}}{2}\left(-\frac{1}{\beta}\sum_{i=1}^{N}\sigma_i^z z_i \partial_{z_i} + \sum_{i<j}\frac{z_i \sigma_j^z - z_j \sigma_i^z}{z_i - z_j}P_{ij} + \frac{1}{2}\sum_{i\neq j}\frac{z_i + z_j}{z_i - z_j}\sigma_j^z\right),
\end{aligned} \quad (109)
$$

since we are interested in the fermionic sector, i.e. the space of vectors on which the action of $s_{ij}$ and $-P_{ij}$ coincide for all $i \neq j$. We recall that $P'$ is the total momentum operator (81), which comes from the standard Calogero–Sutherland-style charges (95), and therefore commute with the transfer matrix and all operators obtained from it. Thus we may drop it to obtain (6). We emphasise that these Heisenberg-style charges commute with the standard Calogero–Sutherland charges coming from the quantum determinant for any value of $\kappa$.

In the untwisted case, the transfer matrix simplifies to

$$
t\left(x + \frac{1}{2}; 1\right) = 2 + N x^{-1} + \left(t_2 - \beta^{-1}P'\right)x^{-2} + \left(t_3 + 2\beta^{-2}H' + E^0\right)x^{-3} + O\left(x^{-4}\right). \quad (110)
$$

Here $t_2 = \sum_{i<j} P_{ij}$, and

$$
\begin{aligned}
t_3 &= \sum_{i<j<k} P_{jk} P_{ij} - \sum_{i<j} (d_i + d_j) P_{ij} \\
&= -\frac{1}{\beta} \sum_{i<j} (z_i \, \partial_i + z_j \, \partial_j) P_{ij} + \sum_{i<j} \sum_{k(\neq i,j)} \left( 1 - \frac{z_i}{z_{ik}} - \frac{z_j}{z_{jk}} \right) P_{ij} \\
&\quad + \sum_{i<j<k} \left[ \left( 2 - \frac{z_i}{z_{ij}} - \frac{z_j}{z_{jk}} - \frac{z_k}{z_{ki}} \right) P_{ij} P_{jk} + \left( 2 - \frac{z_j}{z_{ji}} - \frac{z_k}{z_{kj}} - \frac{z_i}{z_{ik}} \right) P_{jk} P_{ij} \right] \\
&= -\frac{1}{\beta} {\sum_{i,j}}' P_{ij} \, z_i \, \partial_i - \frac{1}{2} {\sum_{i,j,k}}' \frac{z_i + z_k}{z_i - z_k} P_{ij} + \frac{1}{2} {\sum_{i,j,k}}' \left[ \frac{1}{3} - \frac{z_i + z_j}{z_i - z_j} \right] P_{ij} P_{jk} \,,
\end{aligned}
\tag{111}
$$

where we once again replaced $s_{ij}$ with $-P_{ij}$, and in the last line the prime indicates that equal values of summation indices are omitted from the sum. Thus, the Heisenberg-style charges go beyond the standard Calogero–Sutherland-style charges even in the periodic case.

## 4.2 Internal Bethe Ansatz

It remains to diagonalise our Heisenberg-style symmetries by algebraic Bethe Ansatz. Using the decomposition (96), we can restrict ourselves to a spin-Calogero–Sutherland eigenspace $\mathcal{F}_\lambda$ labelled by an allowed partition $\lambda \in \mathcal{P}$. In this subspace, the spectrum of the transfer matrix $t(x;\kappa)$ from (107) coincides with the spectrum of the transfer matrix

$$
t_\lambda(x;\kappa) = \text{Tr}_0 \big[ \kappa^{\sigma_0^z} (T_\lambda)_0(x) \big] ,
\tag{112}
$$

of the effective spin chain, which is just an inhomogeneous Heisenberg spin chain. Therefore we can use the results of Section 2.

We can view the algebraic Bethe Ansatz in three ways. First, inside the effective spin chain $\mathcal{H}_\lambda$ with monodromy (106), the algebraic Bethe Ansatz has the standard form from Section 2,

$$
B_\lambda(x_1) \cdots B_\lambda(x_M) | \uparrow \cdots \uparrow \rangle \in \mathcal{H}_\lambda .
\tag{113}
$$

Second, thinking of the effective spin chain as the Yangian-invariant subspace inside $V_2^{\otimes N}$ with inhomogeneities $\delta_1(\lambda), \ldots, \delta_N(\lambda)$, the algebraic Bethe Ansatz uses the $B$-operator contained in the monodromy matrix (103) and pseudovacuum (104). Third, inside the fermionic eigenspace $\mathcal{F}_\lambda$ we start from $|0_\lambda\rangle$ given by (101), and use the monodromy matrix (93) with Dunkl operators to perform the algebraic Bethe Ansatz,

$$
B(x_1) \cdots B(x_M) |0_\lambda\rangle \in \mathcal{F}_\lambda .
\tag{114}
$$

Since $\mathcal{H}_\lambda$, its image as invariant subspace of $V_2^{\otimes N}$ and $\mathcal{F}_\lambda$ are isomorphic as Yangian modules, the three perspectives are equivalent. We emphasise that the $M$-magnon sector of the effective spin chain $\mathcal{H}_\lambda$ (of length $L_\lambda$) corresponds to $M_\lambda + M$ magnons inside $V_2^{\otimes N}$ and $\mathcal{F}_\lambda$.

According to (25) and (94) the eigenvalue of the transfer matrix $t(x;\kappa)$ on the Bethe vector in $\mathcal{F}_\lambda$ with Bethe roots $(x_1, \ldots, x_M)$ reads

$$
\tau(x;\kappa) = \prod_{j \in J_\lambda} \frac{x + \delta_j(\lambda) + \frac{1}{2}}{x + \delta_j(\lambda) - \frac{1}{2}} \left( \kappa \frac{Q(x-1)}{Q(x)} \prod_{i \in I_\lambda} \frac{x + \delta_i(\lambda) + \frac{1}{2}}{x + \delta_i(\lambda) - \frac{1}{2}} + \kappa^{-1} \frac{Q(x+1)}{Q(x)} \right),
\tag{115}
$$

where the Bethe roots $x_1, \ldots, x_M$ solve the Bethe equations (21), which here read

$$
\kappa^2 \prod_{i \in I_\lambda} \frac{x_m + \delta_i(\lambda) + \frac{1}{2}}{x_m + \delta_i(\lambda) - \frac{1}{2}} = -\frac{Q(x_m + 1)}{Q(x_m - 1)} ,
\tag{116}
$$

with $Q(x) \equiv \prod_{m=1}^{M}(x - x_m)$. For $\kappa^2 \neq 1$ their Wronskian form is the QQ-relation (28), i.e.

$$\kappa Q\left(x - \frac{1}{2}\right)\widetilde{Q}\left(x + \frac{1}{2}\right) - \kappa^{-1}Q\left(x + \frac{1}{2}\right)\widetilde{Q}\left(x - \frac{1}{2}\right) = (\kappa - \kappa^{-1})\prod_{i \in I_\lambda}(x + \delta_i(\lambda)), \qquad (117)$$

for some degree $L_\lambda - M$ polynomial $\widetilde{Q}$. By Section 2.3.3, in the periodic case it instead reads

$$Q\left(x - \frac{1}{2}\right)\widetilde{Q}\left(x + \frac{1}{2}\right) - Q\left(x + \frac{1}{2}\right)\widetilde{Q}\left(x - \frac{1}{2}\right) = (L_\lambda + 1 - 2M)\prod_{i \in I_\lambda}(x + \delta_i(\lambda)), \qquad (118)$$

and $\widetilde{Q}$ has degree $L_\lambda + 1 - M$. The transfer matrix being Hermitian provided $\beta > 0$ and $\kappa \in \mathbb{R}$, its spectrum must be real. Hence, $Q$ is a real polynomial and its roots can either be real or contain complex conjugate pairs. In the conventions of Section 2, the inhomogeneities are imaginary, and the solutions of the Bethe equations here have a very different structure than for the usual (homogeneous) Heisenberg spin chain. In particular, $\{u_1, \ldots, u_M\} = \{i x_1, \ldots, i x_M\}$ is not necessarily stable under complex conjugation (although $\{x_1, \ldots, x_M\}$ is for real $\kappa$). In Sections 4.3.2–4.4 we will give some simple examples of Bethe roots.

Expanding the transfer-matrix eigenvalue (115) around $x \to +\infty$ and comparing with (108) and (110) we obtain the eigenvalues of the conserved charges. In the untwisted case, the eigenvalue of $t_2$ is

$$\tau_2 = \left(\frac{L_\lambda}{2} - M\right)\left(\frac{L_\lambda}{2} - M + 1\right) + \frac{N(N-4)}{4}. \qquad (119)$$

This simply means that the eigenstate is in an irreducible $\mathfrak{sl}_2$-module of spin $\frac{L_\lambda}{2} - M$. The eigenvalue of $t_3$ is

$$\tau_3 = -\left(\frac{L_\lambda}{2} - M + 1\right)\left(2\sum_{m=1}^{M}x_m + \sum_{i \in I_\lambda}\delta_i(\lambda)\right) + \left(2 - \frac{N}{2}\right)\sum_{i=1}^{N}\delta_i(\lambda) + \frac{N-2}{2}\left(\tau_2 - \frac{N(N-1)}{6}\right). \qquad (120)$$

In the twisted case, the transfer matrix eigenvalue behaves as

$$\tau(x;\kappa) = \kappa + \kappa^{-1} + \left[\frac{\kappa + \kappa^{-1}}{2}N + (\kappa - \kappa^{-1})\left(\frac{L_\lambda}{2} - M\right)\right]x^{-1} + \left[\frac{\kappa + \kappa^{-1}}{2}\left(\tau_2 - \sum_{i=1}^{N}\delta_i(\lambda)\right)\right.$$

$$\left. + \frac{\kappa - \kappa^{-1}}{2}\left((N-1)\left(\frac{L_\lambda}{2} - M\right) - 2\sum_{m=1}^{M}x_m - \sum_{i \in I_\lambda}\delta_i(\lambda)\right)\right]x^{-2} + O(x^{-3}). \qquad (121)$$

These are the energies of our Heisenberg-style symmetries.

## 4.3  Limits

To illustrate our construction we consider some limits. As we saw in Subsection 2.3.4, for extreme twist $\kappa \to \infty$ ($\kappa \to 0$), for each irreducible submodule — or, equivalently, effective spin chain — the Bethe states approach the Yangian Gelfand–Tsetlin basis diagonalising the $A$- (respectively $D$-)operator contained in the twisted transfer matrix (107). Let us here study the behaviour of the Bethe roots and the spectrum in two other interesting limits $\beta \to 0$ ($\beta \to +\infty$) of the coupling constant, in which the kinetic energy dominates (is dominated by) the potential energy.

### 4.3.1 Free-fermion limit $\beta \to 0$

When the coupling constant $\beta$ vanishes, the spin-Calogero–Sutherland model becomes a free-fermion model. The rescaled Dunkl operators reduce to the particle momentum operators $\beta d_i \to z_i \partial_{z_i}$ as $\beta \to 0$, and nonsymmetric Jack polynomials boil down to monomials $E_\mu(z) \to z^\mu \equiv z_1^{\lambda_1} \cdots z_N^{\lambda_N}$ (i.e. plane waves), with rescaled eigenvalues $\beta \delta_i(\mu) \to \mu_i$ equal to their degrees (wave numbers) in the $z_i$ ($\mu \in \mathbb{Z}^N$). The spin-Calogero–Sutherland eigenvectors can be described elegantly in terms of the wedge basis [24], see also §2.3.2 in [41].

The solutions to the Bethe equations in $\mathcal{F}_\lambda$ with allowed partition $\lambda \in \mathcal{P}$ are also particularly simple in this limit: the rescaled Bethe roots $\{\beta x_1, \dots, \beta x_M\}$ form a subset of the distinct parts $\{-\lambda_i\}_{i \in \{1,\dots,N\} \setminus J_\lambda}$ of $-\lambda$. The monodromy matrix (13) can be expanded in $\beta$ in the following way:

$$T_0\left(\frac{x}{\beta} + \frac{1}{2}\right) = 1 + \beta \sum_{i=1}^N \frac{P_{0i}}{x + z_i \partial_i} + \beta^2 \left( \sum_{i<j} \frac{P_{0i} P_{0j}}{(x + z_i \partial_i)(x + z_j \partial_j)} - \sum_{i=1}^N \frac{P_{0i} d_i^\circ}{(x + z_i \partial_i)^2} \right) + O(\beta^3). \quad (122)$$

Here $(x + z_i \partial_i)^{-1}$ acts on monomials as $(x + z_i \partial_i)^{-1} z^\lambda = (x + \lambda_i)^{-1} z^\lambda$, and at order $\beta^2$ we picked up a contribution from the subleading part of the Dunkl operator,

$$d_i^\circ \equiv -\sum_{j=1}^{i-1} \frac{z_i}{z_{ji}} (1 + P_{ij}) + \sum_{j=i+1}^N \frac{z_j}{z_{ij}} (1 + P_{ij}) + \frac{N+1-2i}{2}, \quad (123)$$

where we used the fermionic condition to replace $s_{ij}$ by $-P_{ij}$. Hence the transfer matrix is

$$
\begin{aligned}
t\left(\frac{x}{\beta} + \frac{1}{2}; \kappa\right) = {}& \kappa + \kappa^{-1} + \beta \sum_{i=1}^N \frac{\kappa^{\sigma_i^z}}{x + z_i \partial_i} \\
&+ \beta^2 \left( \sum_{i<j} \frac{\kappa^{\sigma_j^z} P_{ij}}{(x + z_i \partial_i)(x + z_j \partial_j)} - \sum_{i=1}^N \frac{\kappa^{\sigma_i^z} d_i^\circ}{(x + z_i \partial_i)^2} \right) + O(\beta^3).
\end{aligned}
\quad (124)
$$

To linear order in $\beta$ the eigenvalues of the transfer matrix are of the form

$$\tau\left(\frac{x}{\beta} + \frac{1}{2}; \kappa\right) = \kappa + \kappa^{-1} + \beta \left( \kappa \sum_{m=1}^M \frac{1}{x + \lambda_{i_m}} + \kappa^{-1} \sum_{i \notin I} \frac{1}{x + \lambda_i} \right) + O(\beta^2). \quad (125)$$

Had we not imposed any (anti)symmetry on the eigenvectors, these values would occur in the spectrum for all $\lambda \in \mathbb{Z}^N$ and $I = \{i_1, \dots, i_M\}$ any subset of $\{1, \dots, N\}$. However, for fermionic eigenvectors only some of these eigenvalues are valid. To see this, it is convenient to start from the exact spectrum at small, but finite $\beta$. Examining the Bethe equations (116) in this limit, one realises that in this limit the inhomogeneities become large and the Bethe roots have to stick to them (up to a finite $\kappa$-dependent shift). For $\lambda \in \mathcal{P}$, the solutions to the Bethe equations can be indexed by $I = \{i_1, \dots, i_M\} \subset I_\lambda$. Solving the Bethe equations perturbatively, one finds that the rescaled Bethe roots are

$$
\begin{aligned}
\beta x_m = {}& -\lambda_{i_m} - \frac{\beta}{2} \left( N + 1 - 2i_m + \frac{\kappa + \kappa^{-1}}{\kappa - \kappa^{-1}} \right) \\
&+ \frac{\beta^2}{(\kappa - \kappa^{-1})^2} \left( \sum_{j \in I_\lambda \setminus I} \frac{1}{\lambda_j - \lambda_{i_m}} - \sum_{j \in I \setminus \{i_m\}} \frac{1}{\lambda_j - \lambda_{i_m}} \right) + O(\beta^3).
\end{aligned}
\quad (126)
$$

This relies on the fact that the inhomogeneities are far away from one another: as we noted, for $i \neq j$ in $I_\lambda$ we have

$$\beta \big( \delta_i(\lambda) - \delta_j(\lambda) \big) = \lambda_i - \lambda_j + \beta(j - i) \longrightarrow \lambda_i - \lambda_j \neq 0, \qquad \beta \to 0. \quad (127)$$

Plugging the values of the Bethe roots into the expression (115) for the transfer matrix eigenvalue, one finds that it simplifies to

$$
\tau\left(\frac{x}{\beta};\kappa\right) = \kappa\,\alpha_{\lambda,I}\left(\frac{x}{\beta}\right) + \kappa^{-1}\alpha_{\lambda,I_\lambda\setminus I}\left(\frac{x}{\beta}\right)
$$
$$
+ \frac{\beta^3}{\kappa - \kappa^{-1}}\sum_{m=1}^{M}\sum_{j\in I_\lambda\setminus I}\frac{1}{(x+\lambda_{i_m})(x+\lambda_j)(\lambda_{i_m}-\lambda_j)} + O(\beta^4), \quad (128)
$$

where

$$
\alpha_{\lambda,I}(x) = \prod_{i\in(I_\lambda\setminus I)\cup J_\lambda}\frac{x+\delta_i(\lambda)+\frac{1}{2}}{x+\delta_i(\lambda)-\frac{1}{2}}, \quad (129)
$$

is an eigenvalue of the element $A(x)$ of the monodromy matrix. The first line can be expanded further in $\beta$. Notice, however, that the transfer-matrix eigenvalues start differing from the sum of the eigenvalues of $\kappa A$ and $\kappa^{-1}D$ only at order $\beta^3$.

Finally observe that in the infinite twist limit when $\kappa\to+\infty$, the Bethe roots become equal to $-\delta_{i_m}(\lambda)-1/2+O(\beta^2)$ while the transfer matrix eigenvalue becomes $\kappa\,\alpha_{\lambda,I}(\beta^{-1}x)+O(\beta^4)$. As discussed in Section 2.3.4, these should actually be the exact values at all orders in $\beta$, when $\kappa\to+\infty$. A similar observation can be made for the limit $\kappa\to 0$.

### 4.3.2 Strong-coupling limit $\beta\to\infty$ and the Haldane–Shastry spin chain

Now consider the opposite limit, $\beta\to\infty$, which is dominated by the potential energy. In this limit some of the spaces $\mathcal{F}_\lambda\cong\mathcal{H}_\lambda$ turn into reducible, indecomposable representations of the Yangian. This is because the differences between eigenvalues (78) of the Dunkl operators become integer-valued: when $\lambda$ is a partition, one has

$$
\delta(\lambda)\longrightarrow\frac{1}{2}(N-1,N-3,\dots,1-N), \qquad \beta\to\infty. \quad (130)
$$

From Section 2.4.4 we know that here *all* pairs $j, j+1$ of neighbouring sites are fused into singlets, leading to many invariant subspaces, and that at the same time the algebraic Bethe Ansatz allows us to generate all eigenstates in $\mathcal{F}_\lambda$. By taking the limit $\beta\to\infty$ of the equations (115)–(117) one finds the corresponding transfer-matrix eigenvalues and the Bethe roots. This is the strong-coupling limit of the spin-Calogero–Sutherland model. We are most interested in going one step further and reducing the infinite-dimensional space of states to a finite-dimensional Hilbert space.

In the freezing procedure we supplement the strong-coupling limit $\beta\to\infty$ by applying

$$
\mathrm{ev}\colon (z_1,\dots,z_N)\longmapsto\left(1, e^{\frac{2i\pi}{N}},\dots, e^{\frac{2i(N-1)\pi}{N}}\right), \quad (131)
$$

to evaluate all functions of the $z_j$ at consecutive $N$th roots of unity. Then the (fermionic) Calogero–Sutherland Hamiltonian reduces to that of the (antiferromagnetic) Haldane–Shastry spin chain [36, 38, 41],

$$
\beta^{-1}\widetilde{H}'\to H^{\mathrm{HS}} = \sum_{i<j}\frac{1+P_{ij}}{4\sin^2\left[\frac{\pi}{N}(i-j)\right]}. \quad (132)
$$

In the freezing limit most of the eigenvectors vanish. We describe the result without proofs, which will be given in a separate publication. If $\lambda_1-\lambda_N\geqslant N$ then the evaluation projects $\mathcal{F}_\lambda$ to $\{0\}$. Otherwise, the evaluation of $\mathcal{F}_\lambda$ is non-trivial and completely determined by the motif $J_\lambda$. It is described as follows: in the limit $\beta\to\infty$, the inhomogeneities of the effective spin chain are (130) where for each $j\in J_\lambda$ the $j$ and $j+1$st elements are dropped. This means that the inhomogeneities can be separated into (maximal) groups of consecutive half-integers decreasing

in steps of 1 within each group. Each such group of $p$ successive inhomogeneities (or '$p$-string') corresponds to a copy of the spin-$p/2$ representation $V_{p+1}$ of $\mathfrak{sl}_2$ (see §12.1.E in [62]). After evaluation, if nonzero, the space $\mathcal{F}_\lambda$ is isomorphic to the product of all these $V_{p+1}$ [38, 55]. The freezing procedure actually amounts to quotienting out all the invariant subspaces. We emphasise that the resulting quotient space only depends on the motif $J_\lambda$ recording the repeats in $\lambda$, and is insensitive to the precise values $\lambda_i$ that occur. For instance, if $N = 6$ and we start from the motif $J_\lambda = \{4\}$, then the evaluation of $\mathcal{F}_\lambda$ will be isomorphic to $V_4 \otimes V_2$. Similarly, if $N = 11$, the motif $\{2, 5\}$ will correspond to a subspace isomorphic to $V_2 \otimes V_3 \otimes V_5$. The Calogero–Sutherland eigenvectors that survive evaluation become eigenvectors of the Haldane–Shastry spin chain. For highest-weight vectors the result can be described in terms of a symmetric Jack polynomial in $M$ variables at the (zonal spherical) point $\alpha = 1/2$ [38], see also [41].

In the freezing limit the derivatives in $z_i$ in our Heisenberg-style symmetries disappear. The twisted charge (109) becomes

$$t_2(\kappa) \;\to\; t_2^{\mathrm{HS}}(\kappa) = \frac{\kappa + \kappa^{-1}}{2} \sum_{i<j} P_{ij} + \frac{\kappa - \kappa^{-1}}{4\,\mathrm{i}} \sum_{i<j} \frac{\mathrm{e}^{\mathrm{i}\pi(i-j)/N}\sigma_j^z - \mathrm{e}^{\mathrm{i}\pi(j-i)/N}\sigma_i^z}{\sin[\frac{\pi}{N}(i-j)]} P_{ij}\,, \qquad (133)$$

while the periodic charge (111) yields

$$\begin{aligned}
t_3^{\mathrm{HS}} &= \frac{1}{2} \sum_{i<j<k} \Big[ P_{ij}\,P_{jk} + P_{jk}\,P_{ij} \\
&\qquad\qquad + \mathrm{i}\big(\cot\big[\tfrac{\pi}{N}(i-j)\big] + \cot\big[\tfrac{\pi}{N}(j-k)\big] + \cot\big[\tfrac{\pi}{N}(k-i)\big]\big)\big(P_{ij}\,P_{jk} - P_{jk}\,P_{ij}\big) \Big] \\
&= \frac{1}{2} {\sum_{i,j,k}}' \left(\frac{1}{3} + \mathrm{i}\cot\big[\tfrac{\pi}{N}(i-j)\big]\right) P_{ij}\,P_{jk}\,, \qquad\qquad\qquad (134)
\end{aligned}$$

where once again the prime indicates that equal indices are omitted from the sum. These operators can be viewed as refinements of the standard hierarchy of the Haldane–Shastry spin chain [38, 40], as they commute with each other and, even in the twisted case, with the (periodic) spin-chain translation operator. Moreover, (133) commutes with $S^z$, and (134) is $\mathfrak{sl}_2$ invariant; but, unlike the Hamiltonian (132), neither commutes with the Yangian. Note that the periodic charge (134) is similar to Inozemtsev's charge (4). Our approach thus reconciles the latter with the standard approach to the Haldane–Shastry spin chain based on Yangian symmetry [38, 40], while providing a systematic way to obtain higher Heisenberg-style charges, depending on an additional arbitrary twist parameter $\kappa$.

The spectrum of our Heisenberg-style symmetries, e.g. (120), is determined by the transfer-matrix eigenvalue (115) once one solves the Bethe equations (117) or (118). Only those solutions for which $Q$ and $\widetilde{Q}$ have no common root will correspond to eigenvectors that do not belong to an invariant subspace (cf. Appendix B), and hence survive freezing. Examples of explicit sets of Bethe roots for $\kappa = 1$ are

$$N = 7\,, \quad J_\lambda = \{4\}\,, \quad M = 2: \qquad \{x_1, x_2\} = \left\{1 - \mathrm{i}\frac{\sqrt{3}}{2}, 1 + \mathrm{i}\frac{\sqrt{3}}{2}\right\}\,, \qquad (135)$$

and

$$N = 8\,, \quad J_\lambda = \{4\}\,, \quad M = 3: \qquad \{x_1, x_2, x_3\} = \left\{-\mathrm{i}\sqrt{5}, 0, \mathrm{i}\sqrt{5}\right\}\,. \qquad (136)$$

The resulting Bethe vectors provide a new eigenbasis for the Haldane–Shastry spin chain that reduce to the Gelfand–Tsetlin basis in the limit of extreme twist.

## 4.4 Example: $N = 4$

We illustrate our constructions in an example where we can explicitly build the Bethe-Ansatz eigenvectors of our Heisenberg-style symmetries such as (109) or (111). We start with the spin-Calogero–Sutherland model, and then turn to the Haldane–Shastry spin chain by freezing.

We consider the case of $N = 4$ particles. We focus on the partition $\boldsymbol{\lambda} = (2,1,1,0)$, with motif $J_{\boldsymbol{\lambda}} = \{2\}$. By (85) the momentum and energy are $P'(\boldsymbol{\lambda}) = 4$ and $E'(\boldsymbol{\lambda}) = 3(1+\beta)$. Let us construct our Bethe-Ansatz basis for $\mathcal{F}_{\boldsymbol{\lambda}}$. The corresponding effective spin chain has length $L_{\boldsymbol{\lambda}} = 2$. In the periodic case $\kappa = \pm 1$ the Bethe states are determined by the highest-weight state in $\mathcal{F}_{\boldsymbol{\lambda}}$ together with $\mathfrak{sl}_2$ symmetry, but for general twist the two states at the equator (having one magnon in the language of the effective spin chain) are nontrivial, and it is this case we will focus on.

The highest-weight vector inside $\mathcal{F}_{(2,1,1,0)}$ occurs at $M = 1$ and is of the form (101), i.e.

$$
\begin{aligned}
|0_{(2,1,1,0)}\rangle = {}& f(z_1; z_2, z_3, z_4)\,|\downarrow\uparrow\uparrow\uparrow\rangle - f(z_2; z_1, z_3, z_4)\,|\uparrow\downarrow\uparrow\uparrow\rangle \\
& + f(z_3; z_1, z_2, z_4)\,|\uparrow\uparrow\downarrow\uparrow\rangle - f(z_4; z_1, z_2, z_3)\,|\uparrow\uparrow\uparrow\downarrow\rangle ,
\end{aligned}
\qquad f \equiv f_{(2,1,1,0)}, \qquad (137)
$$

because it should be totally antisymmetric. For the same reason, the polynomial $f$ must be antisymmetric in the last three variables, $f = -s_{23}f = -s_{34}f$. This does not allow equal exponents for $z_2, z_3, z_4$, so for $\boldsymbol{\lambda} = (2,1,1,0)$ we have $f = z_1 z_2^2 z_3 + $ lower, where the remaining terms are lower in the dominance order. The partial antisymmetry then requires a partial Vandermonde factor $(z_2 - z_3)(z_2 - z_4)(z_3 - z_4) = z_2^2 z_3 +$ lower, which fixes the remaining symmetric part as [18]

$$
\begin{aligned}
f(z_1; z_2, z_3, z_4) &= z_1(z_2 - z_3)(z_2 - z_4)(z_3 - z_4) \\
&= -(1 - s_{23} - s_{34} + s_{23} s_{34} + s_{34} s_{23} - s_{24})\, E_{(1,0,1,2)} ,
\end{aligned}
\qquad (138)
$$

in accordance with (101).

Having constructed the vacuum state, we now need to solve the twisted Bethe equations. The eigenvalues of the Dunkl operators are read off from (102) as

$$
\delta_1 = \frac{2}{\beta} + \frac{3}{2}, \qquad \delta_2 = \frac{1}{\beta} + \frac{1}{2}, \qquad \delta_3 = \frac{1}{\beta} - \frac{1}{2}, \qquad \delta_4 = -\frac{3}{2}. \qquad (139)
$$

Out of these, only $\delta_1$ and $\delta_4$ enter the Bethe equations (116) since $I_{\boldsymbol{\lambda}} = \{1,4\}$. As explained above we are interested in the 1-magnon states. We find the following two values of the Bethe root:

$$
x_{1,\pm} = -\frac{(\beta + 2)\kappa + (\beta - 2)\kappa^{-1} \pm \kappa^{-1}\sqrt{(3\beta + 2)^2 \kappa^4 - 2(7\beta^2 + 12\beta + 4)\kappa^2 + (3\beta + 2)^2}}{2\beta(\kappa - \kappa^{-1})}.
\qquad (140)
$$

Note that the expansion of (140) for $\beta \to 0$ matches (126) that we obtained in the free fermion limit.

Now we consider the freezing limit as described in the previous subsection. If we evaluate $|0_{(2,1,1,0)}\rangle$ using ev : $(z_1, z_2, z_3, z_4) \mapsto (1, i, -1, -i)$, we obtain a Yangian-highest-weight eigenvector of the Haldane–Shastry spin chain:

$$
\begin{aligned}
\mathrm{ev}\big[|0_{(2,1,1,0)}\rangle\big] &= -4\,i\,\big[|\downarrow\uparrow\uparrow\uparrow\rangle - |\uparrow\downarrow\uparrow\uparrow\rangle + |\uparrow\uparrow\downarrow\uparrow\rangle - |\uparrow\uparrow\uparrow\downarrow\rangle\big] \\
&= -4\,i\,\sum_{i=1}^{4} \mathrm{ev}\big[P_{(2)}^{\star}(z_i)\big]|i\rangle\rangle , \qquad P_{(2)}^{\star}(z) = z^2 ,
\end{aligned}
\qquad (141)
$$

---

[18]As $\bar{\boldsymbol{\lambda}} = (1,0,1,2)$ is the *lowest* amongst all reorderings of $\boldsymbol{\lambda}$ with $\bar{\lambda}_1 = 1$, $E_{\bar{\boldsymbol{\lambda}}}$ is the *simplest* amongst the corresponding nonsymmetric Jack polynomials. Explicitly, $E_{\bar{\boldsymbol{\lambda}}} = z_1 z_3 z_4^2 + \frac{\beta}{2\beta+1}(z_2 z_3 z_4^2 + z_1 z_2 z_3 z_4)$.

where the second line contains the case $M = 1$ of the standard Haldane–Shastry (Yangian highest-weight) wave function $\text{Vand}(z_1,\dots,z_M)^2 P_\nu^\star(z_1,\dots,z_M)$ with $P_\nu^\star(\boldsymbol{z})$ a Jack polynomial at $\alpha^\star = 1/2$. The vector (141) is just a magnon with (lattice) momentum $p = \pi$. Note that it is *not* the same as the highest-weight vector (104) of the effective spin chain embedded in the 'ambient' space $V_2^{\otimes N}$ with special inhomogeneities, even though the latter has the same dimension as the Hilbert space of the Haldane–Shastry spin chain.

The Bethe roots in the freezing limit are found from their original values (140) by taking $\beta \to \infty$, which gives

$$x_{1,\pm}^\circ = -\frac{\kappa + \kappa^{-1} \pm \kappa^{-1}\sqrt{9\kappa^4 - 14\kappa^2 + 9}}{2(\kappa - \kappa^{-1})}\,. \tag{142}$$

Writing $B^\circ(x) \equiv \lim_{\beta\to\infty} B(x)$, we obtain the Bethe states by acting on the vacuum with the B-operator. We find

$$\text{ev}\Big[B^\circ(x_{1,\pm}^\circ)\,|0_{(2,1,1,0)}\rangle\Big] = c_\pm\,|\Psi_\pm\rangle\,, \quad c_\pm = \pm 2\sqrt{2}\,\mathrm{i}\left(1 \pm \kappa^{-1}\right)\left(\frac{(1 - \kappa^{-1})(x_{1,-}^\circ - 2)}{\sqrt{2}\,x_{1,+}^\circ}\right)^{\pm 1}, \tag{143}$$

where the two linearly independent one-magnon eigenstates read

$$|\Psi_\pm\rangle = \mathrm{i}\left(|\uparrow\uparrow\downarrow\downarrow\rangle - |\uparrow\downarrow\downarrow\uparrow\rangle - |\downarrow\uparrow\uparrow\downarrow\rangle + |\downarrow\downarrow\uparrow\uparrow\rangle\right) - x_{1,\pm}^\circ\left(|\downarrow\uparrow\downarrow\uparrow\rangle - |\uparrow\downarrow\uparrow\downarrow\rangle\right). \tag{144}$$

The respective eigenvalues of the operator $t_2^{\text{HS}}$ from (133) are $-(\kappa - \kappa^{-1})\,x_{1,\pm}^\circ$ in accordance with the coefficient of $x^{-2}$ in equation (121). These are nontrivial eigenvectors for the Haldane–Shastry chain with motif $\{2\}$ that moreover are eigenvectors of our Heisenberg-style symmetries for any twist $\kappa$.

In the periodic limit we obtain $x_{1,+} \to \pm\infty$ depending on whether $\kappa \to 1^\pm$ from above or below. In this case $B(x) \sim x\,S^-$ creates a descendant, reflecting the $\mathfrak{sl}_2$-symmetry in this limit. The other Bethe root becomes $x_{1,-} \to -\beta^{-1}$. Since in this case all other vectors at $M = 2$ are either $\mathfrak{sl}_2$-descendants or have (Yangian) highest weight, the corresponding Bethe vector can alternatively be determined by orthogonality at $\kappa = 1$. Further taking the freezing limit gives $x_{1,-}^\circ = 0$. The resulting vectors match the $\kappa \to 1$ limit of (143),

$$\begin{aligned}
\frac{\mathrm{i}}{4}\lim_{\kappa\to 1}\frac{1}{\kappa - \kappa^{-1}}\,c_+\,|\Psi_+\rangle &= -\big(|\downarrow\uparrow\downarrow\uparrow\rangle - |\uparrow\downarrow\uparrow\downarrow\rangle\big) = -\frac{\mathrm{i}}{8}\,S^-\,\text{ev}\big[|0_{(2,1,1,0)}\rangle\big]\,,\\
\frac{\mathrm{i}}{4}\lim_{\kappa\to 1}(\kappa - \kappa^{-1})\,c_-\,|\Psi_-\rangle &= \mathrm{i}\left(|\uparrow\uparrow\downarrow\downarrow\rangle - |\uparrow\downarrow\downarrow\uparrow\rangle - |\downarrow\uparrow\uparrow\downarrow\rangle + |\downarrow\downarrow\uparrow\uparrow\rangle\right).
\end{aligned} \tag{145}$$

As expected, the former is the $\mathfrak{sl}_2$-descendant of the $p = \pi$ magnon, while the latter has highest weight for $\mathfrak{sl}_2$.

## 5 Conclusion

In this paper we showed how the commuting family of spin-Calogero–Sutherland Hamiltonaians can be refined using a transfer matrix. This gives new Heisenberg-style symmetries as well as a new Bethe-Ansatz eigenbasis for the spin-Calogero–Sutherland model. Along the way we reviewed and explored nontrivial features of the spin chains arising in this construction, which involve fusion. One salient feature is the description of the Yangian highest-weight vector in the invariant subspace for singlet fusion in algebraic Bethe-Ansatz form, using $B$-operators at special fixed Bethe roots. Via freezing, our results also provide a new Bethe-Ansatz eigenbasis for the Haldane–Shastry chain. We illustrate our framework in several special cases, including

its reduction to the Yangian Gelfand–Tsetlin basis in the limit of extreme twist, and a number of nontrivial examples for small system size.

There are several interesting directions left for the future.

- Following [41, 69] we considered the fermionic spin-Calogero–Sutherland model. One can analogously use a Bethe-Ansatz analysis in the bosonic case, which was also studied by Takemura–Uglov [23].

- Our results should naturally generalise to higher-rank $\mathfrak{sl}_r$ spin-Calogero–Sutherland models beyond the case $r = 2$ considered here. We also expect that our construction can be extended to xxz-type models, i.e. the spin-Ruijsenaars–Macdonald model as well as the $q$-deformed Haldane–Shastry spin chain [38, 54, 55, 70].

- Another interesting direction is extending our results to other Yangian-invariant spin chains, like the (rational) Polychronakos–Frahm model [36, 71] or the (hyperbolic) Frahm–Inozemtsev system [72].

- We plan to expand on and prove our claims from Section 4.3.2 about freezing at the level of the eigenvectors and representation theory (in particular, the differences between bosonic and fermionic cases reflected in different kinds of fusion).

- Finally, our Heisenberg-style symmetries provide a promising arena to develop Sklyanin's separation of variables (SoV) [73] for long-range models with spins. A key motivation for this comes from integrability in gauge/string (AdS/CFT) duality, where long-range spin chains feature prominently [1, 2], and SoV methods are starting to bring about powerful new results [74–77]. The advantages of our Hamiltonians include the presence of true long-range interactions (unlike in the standard Heisenberg chains), absence of Yangian symmetry (unlike in the spin-Calogero–Sutherland model or Haldane–Shastry chain), and availability of all standard algebraic tools (unlike in models such as the Inozemtsev chain). In combination with recent progress in SoV for higher rank models, see e.g. [59, 78–81], SoV methods for long-range systems should help to develop new ways for computing correlators in AdS/CFT and might shed further light on the mathematical structures behind SoV in general.

## Acknowledgments

We thank V. Pasquier for discussions. JL thanks A. Ben Moussa for discussing fusion and short exact sequences, and K. Takemura for interest and discussions.

**Funding information** GF is grateful to the Azrieli Foundation for the award of an Azrieli Fellowship. The work of JL was funded by LabEx Mathématique Hadamard (LMH), and in the final stage by ERC-2021-CoG – BrokenSymmetries 101044226. Part of this work was carried out during the stay of three of us (JL, FLM and DS) at the NCCR SwissMAP workshop *Integrability in Condensed Matter Physics and QFT* (3–12 February 2023) at the SwissMAP Research Station in Les Diablerets. These authors would like to thank the Swiss National Science Foundation, which funds SwissMAP (grant number 205607) and, in addition, supported the event via the grant IZSEZ0_215085.

## A Fused $R$-matrix

To see explicitly what is happening with the monodromy matrix when $\theta_{j+1} = \theta_j \mp \mathrm{i}$ we focus on the factors $\bar{R}_{0j}(u-\theta_j-\mathrm{i}/2)\bar{R}_{0,j+1}(u-\theta_{j+1}-\mathrm{i}/2)$ in $\bar{T}_0(u)$. It suffices to consider the factors $V_2^{\otimes 3}$ of $V_2 \otimes \mathcal{H}$ corresponding to the auxiliary space and sites $j, j+1$. We are interested in the operator (44) at $\theta_{j+1} = \theta_j \mp \mathrm{i}$. Let us remove the factor $2\mathrm{i}$ coming from (48) and renormalise the $R$-matrix to $\underline{R}(u) \equiv \bar{R}(u)/(u+\mathrm{i})$, which obeys the unitarity condition $\underline{R}_{12}(u)\underline{R}_{21}(-u) = 1$ and initial condition $\underline{R}(0) = P$. We further multiply from the left and right by $\Pi^\pm_{j+1,j}$ and write $u_0 \equiv (\theta_j + \theta_{j+1})/2$. Thus we consider the operator

$$\Pi^\mp_{j+1,j}\,\underline{R}_{0j}(u-u_0)\underline{R}_{0,j+1}(u-u_0+\mathrm{i})\,\Pi^\mp_{j+1,j} = \Pi^\mp_{j+1,j}\,\underline{R}_{0j}(u-u_0+\mathrm{i})\underline{R}_{0,j+1}(u-u_0)\,\Pi^\mp_{j+1,j}\,. \tag{A.1}$$

This equation tells us that the projection of $\bar{R}_{0j}(u-\theta_j-\mathrm{i}/2)\bar{R}_{0,j+1}(u-\theta_{j+1}-\mathrm{i}/2)$ to the copy of $V_1$ or $V_3$ in $V_2 \otimes V_2$ inside $\mathcal{H}$ does not depend on the sign in $\theta_{j+1} = \theta_j \mp \mathrm{i}$.

Consider the basis $|1,1\rangle \equiv |\uparrow\uparrow\rangle$, $|1,0\rangle \equiv (|\uparrow\downarrow\rangle + |\downarrow\uparrow\rangle)/\sqrt{2}$, $|1,-1\rangle \equiv |\downarrow\downarrow\rangle$ for the copy of $V_3$ and $|0,0\rangle \equiv (|\uparrow\downarrow\rangle - |\downarrow\uparrow\rangle)/\sqrt{2}$ for the copy of $V_1$ in $V_2 \otimes V_2$ from $\mathcal{H}$.

The projection of the operator in (A.1) to $V_3$ is equal to $\underline{R}_{0j}^{(1/2,1)}(u-u_0-\mathrm{i}/2)$, where

$$\underline{R}^{(1/2,1)}(u) = \begin{pmatrix} 1 & 0 & 0 & 0 & 0 & 0 \\ 0 & \frac{u+\mathrm{i}/2}{u+3\mathrm{i}/2} & 0 & \frac{\sqrt{2}\,\mathrm{i}}{u+3\mathrm{i}/2} & 0 & 0 \\ 0 & 0 & \frac{u-\mathrm{i}/2}{u+3\mathrm{i}/2} & 0 & \frac{\sqrt{2}\,\mathrm{i}}{u+3\mathrm{i}/2} & 0 \\ 0 & \frac{\sqrt{2}\,\mathrm{i}}{u+3\mathrm{i}/2} & 0 & \frac{u-\mathrm{i}/2}{u+3\mathrm{i}/2} & 0 & 0 \\ 0 & 0 & \frac{\sqrt{2}\,\mathrm{i}}{u+3\mathrm{i}/2} & 0 & \frac{u+\mathrm{i}/2}{u+3\mathrm{i}/2} & 0 \\ 0 & 0 & 0 & 0 & 0 & 1 \end{pmatrix} \quad \text{on} \quad V_2 \otimes V_3 \qquad \text{(A.2)}$$

is the $R$-matrix with spin-$1/2$ in the auxiliary space and spin 1 at site $j$ [82] with respect to the basis $(|\uparrow\rangle \otimes |1,1\rangle, |\uparrow\rangle \otimes |1,0\rangle, |\uparrow\rangle \otimes |1,-1\rangle, |\downarrow\rangle \otimes |1,1\rangle, |\downarrow\rangle \otimes |1,0\rangle, |\downarrow\rangle \otimes |1,-1\rangle)$ of $V_2 \otimes V_3 \subset V_2^{\otimes 3}$.

The projection to $V_1$ is $\underline{R}_{0j}^{(1/2,0)}(u-u_0-\mathrm{i}/2)$ where[19]

$$\underline{R}^{(1/2,0)}(u) = \begin{pmatrix} 1 & 0 \\ 0 & 1 \end{pmatrix} \cdot \mathrm{qdet}\,\underline{R}(u), \quad \text{on} \quad V_2 \otimes V_1\,, \qquad \mathrm{qdet}\,\underline{R}(u) = \frac{u-\mathrm{i}/2}{u+\mathrm{i}/2}\,. \tag{A.3}$$

## B On the derivation of the Bethe equations with fusion

### B.1 Derivation from the QQ-relation

Let us discuss a subtlety in the presence of fusion in the derivation of Bethe equations in the form (21) from the QQ-relation (28), i.e. $\left(\kappa - \kappa^{-1}\right)\bar{Q}_\theta = \bar{Q}^-\widetilde{\bar{Q}}^+ - \bar{Q}^+\widetilde{\bar{Q}}^-$. Shifting the argument to $u \to u+\mathrm{i}/2$ or $u \to u-\mathrm{i}/2$ gives

$$\left(\kappa - \kappa^{-1}\right)\bar{Q}_\theta^+ = \bar{Q}\,\widetilde{\bar{Q}}^{++} - \bar{Q}^{++}\widetilde{\bar{Q}}\,, \qquad \left(\kappa - \kappa^{-1}\right)\bar{Q}_\theta^- = \bar{Q}^{--}\widetilde{\bar{Q}} - \bar{Q}\,\widetilde{\bar{Q}}^{--}\,. \tag{B.1}$$

---

[19]The *quantum determinant* of the Yangian is the central element obtained by singlet fusion in auxiliary space:

$$\mathrm{qdet}_0\bar{T}_0(u+\mathrm{i}/2) \equiv \Pi^-_{00'}\,\bar{T}_0(u+\mathrm{i})\,\bar{T}_{0'}(u) = \bar{A}(u+\mathrm{i})\bar{D}(u) - \bar{B}(u+\mathrm{i})\bar{C}(u) = \bar{D}(u+\mathrm{i})\bar{A}(u) - \bar{C}(u+\mathrm{i})\bar{B}(u)$$

$$= \bar{T}_0(u)\bar{T}_{0'}(u+\mathrm{i})\Pi^-_{00'} = \bar{A}(u)\bar{D}(u+\mathrm{i}) - \bar{C}(u)\bar{B}(u+\mathrm{i}) = \bar{D}(u)\bar{A}(u+\mathrm{i}) - \bar{B}(u)\bar{C}(u+\mathrm{i})\,.$$

For $L=1$ this yields $(u-\theta_1'-\mathrm{i})(u-\theta_1'+\mathrm{i})$ (times the identity), or (A.3) for the normalised $R$-matrix $\underline{R}(u)$.

Evaluating both equations at $u = u_m$ a root of $\bar{Q}$, on the right-hand sides the terms with $\widetilde{\bar{Q}}^{\pm\pm}$ vanish. For generic inhomogeneities, eliminating the remaining $\widetilde{\bar{Q}}$ using the second equation yields the usual Bethe equations (26).

Now consider the fusion of two sites, with $\theta_j = \theta_{j+1} \pm i$ as in Section 2.4.2. Then there is a class of solutions for which $\bar{Q}$ and $\widetilde{\bar{Q}}$ have a common root at

$$u_0 = \frac{\theta_j + \theta_{j+1}}{2}\,. \tag{B.2}$$

Thus all terms in (B.1) vanish separately at $u = u_0$, and we cannot cancel $\widetilde{\bar{Q}}(u_0)$ like before. Instead removing the common factor $u - u_0$ from $\bar{Q}$ and $\widetilde{\bar{Q}}$ gives proper non-singular Bethe equations for an 'effective' spin chain of length $L - 2$ as discussed in Section 2.4.2.

## B.2 Derivation from the algebraic Bethe Ansatz

Here we show how to prove the construction of eigenstates in the form (19) for the case when we fuse two sites by taking, say, $\theta_{j+1} = \theta_j \pm i$ as in Section 2.4.2. The subtlety is that for states in the invariant subspace $V_{\text{inv}} = \Pi^{\mp}_{j,j+1}(\mathcal{H})$ discussed there, some sets of Bethe roots involve the 'frozen' root $u_0 = \theta_j + i/2$. For fusion into singlet, solutions including $u_0$ describe the states in the invariant subspace. These solutions are easily missed when simplifying the Bethe equations (21) to (55). The reason for the existence of such solutions is different for fusion into triplet and singlet.

For $\theta_{j+1} = \theta_j - i$ (fusion into triplet) $\bar{B}(u_0)\,|0\rangle$ vanishes as we show in Appendix C just below, so it cannot be an eigenvector. If instead $\theta_{j+1} = \theta_j + i$ (fusion into singlet), $\bar{B}(u_0)\,|0\rangle$ is nonzero. Let us show that it is an eigenstate of the transfer matrix $\bar{t}(u;\kappa) = \kappa\,\bar{A}(u) + \kappa^{-1}\,\bar{D}(u)$ for any $u$. The standard proof of the algebraic Bethe Ansatz hinges on the commutation relations

$$\bar{A}(u)\bar{B}(u_0) = \frac{u - u_0 - i}{u - u_0}\,\bar{B}(u_0)\bar{A}(u) + \frac{i}{u - u_0}\,\bar{B}(u)\bar{A}(u_0)\,, \tag{B.3}$$

$$\bar{D}(u)\bar{B}(u_0) = \frac{u - u_0 + i}{u - u_0}\,\bar{B}(u_0)\bar{D}(u) - \frac{i}{u - u_0}\,\bar{B}(u)\bar{D}(u_0)\,. \tag{B.4}$$

On $|0\rangle$ the $A$- and $D$-operators can be replaced by their eigenvalues

$$\bar{A}(u)\,|0\rangle = \bar{Q}^+_\theta\,|0\rangle\,, \quad \bar{D}(u)\,|0\rangle = \bar{Q}^-_\theta\,|0\rangle\,. \tag{B.5}$$

Usually the terms with $\bar{B}(u)$ in (B.3) and (B.4) contribute to the 'unwanted' terms, which cancel against each other by virtue of the Bethe equations. However, when $\theta_{j+1} = \theta_j + i$ then $\bar{Q}^\pm_\theta$ both vanish at $u = u_0$, so the 'unwanted' terms cancel *separately*. Hence $\bar{B}(u_0)\,|0\rangle$ is an eigenstate even though the root $u_0$ is not visible in the usual Bethe equations.

## C Action of the *B*-operator at the fixed root

Direct computation shows that

$$\bar{B}(u)\,|0\rangle = i\sum_{i=1}^{L}\prod_{j=1}^{i-1}\left(u - \theta_j + \frac{i}{2}\right)\prod_{j=i+1}^{L}\left(u - \theta_j - \frac{i}{2}\right)|i\rangle\rangle\,. \tag{C.1}$$

For generic values of the inhomogeneities this vector spans the sector with $M = 1$ magnon.

When $\theta_{j+1} = \theta_j + i$ (fusion into singlet) all coefficients with $i \neq j, j+1$ in (C.1) contain a factor $u - (\theta_j + i/2)$, so they vanish at $u = u_0 = (\theta_j + \theta_{j+1})/2 = \theta_j + i/2$. The two remaining coefficients, with $i = j, j+1$, differ by a sign. Thus $|0'\rangle = \bar{B}(u_0)|0\rangle \in \Pi^-_{j,j+1}(\mathcal{H})$ in this case. If instead $\theta_{j+1} = \theta_j - i$ (fusion into triplet) then all coefficients in (C.1) contain a factor $u - (\theta_j - i/2)$. Thus $\bar{B}(u_0)|0\rangle$ now vanishes at $u = u_0 = (\theta_j + \theta_{j+1})/2 = \theta_j - i/2$.

## D   Examples of fusion for low length

### D.1   Generic case and fusion for $L = 2$

Let us illustrate in detail how fusion works for a spin chain with $L = 2$ sites. As representation of $\mathfrak{sl}_2$, which is part of the Yangian, the Hilbert space $\mathcal{H} = V_2 \otimes V_2$ decomposes into the triplet and singlet, $\mathcal{H} = V_3 \oplus V_1$. Pick orthonormal bases $|1,1\rangle \equiv |\uparrow\uparrow\rangle$, $|1,0\rangle \equiv (|\uparrow\downarrow\rangle + |\downarrow\uparrow\rangle)/\sqrt{2}$, $|1,-1\rangle \equiv |\downarrow\downarrow\rangle$ for the copy of $V_3$ and $|0,0\rangle \equiv (|\uparrow\downarrow\rangle - |\downarrow\uparrow\rangle)/\sqrt{2}$ for the copy of $V_1$ in $V_2 \otimes V_2$. In the (not standard!) basis $(|1,1\rangle, |1,0\rangle, |0,0\rangle, |1,-1\rangle)$ we have

$$\bar{A}(u) = \begin{pmatrix} \bar{Q}^+_\theta & 0 & 0 & 0 \\ 0 & \bar{Q}_\theta - \frac{1}{4} & \frac{i}{2}(\theta_1 - \theta_2 + i) & 0 \\ 0 & \frac{i}{2}(\theta_1 - \theta_2 - i) & \bar{Q}_\theta + \frac{3}{4} & 0 \\ 0 & 0 & 0 & \bar{Q}^-_\theta \end{pmatrix}, \tag{D.1}$$

$$\bar{B}(u) = \frac{i}{\sqrt{2}} \begin{pmatrix} 0 & 0 & 0 & 0 \\ 2(u-u_0) & 0 & 0 & 0 \\ -(\theta_1 - \theta_2 - i) & 0 & 0 & 0 \\ 0 & 2(u-u_0) & \theta_1 - \theta_2 + i & 0 \end{pmatrix}, \tag{D.2}$$

$$\bar{C}(u) = \frac{i}{\sqrt{2}} \begin{pmatrix} 0 & 2(u-u_0) & -(\theta_1 - \theta_2 + i) & 0 \\ 0 & 0 & 0 & 2(u-u_0) \\ 0 & 0 & 0 & \theta_1 - \theta_2 - i \\ 0 & 0 & 0 & 0 \end{pmatrix}, \tag{D.3}$$

$$\bar{D}(u) = \begin{pmatrix} \bar{Q}^-_\theta & 0 & 0 & 0 \\ 0 & \bar{Q}_\theta - \frac{1}{4} & -\frac{i}{2}(\theta_1 - \theta_2 + i) & 0 \\ 0 & -\frac{i}{2}(\theta_1 - \theta_2 - i) & \bar{Q}_\theta + \frac{3}{4} & 0 \\ 0 & 0 & 0 & \bar{Q}^+_\theta \end{pmatrix}, \tag{D.4}$$

where $\bar{Q}_\theta = (u - \theta_1)(u - \theta_2)$, $\bar{Q}^\pm_\theta = \bar{Q}_\theta(u \pm i/2)$ and $u_0 = (\theta_1 + \theta_2)/2$. The twisted transfer matrix $\bar{t}(u; \kappa) = \kappa \bar{A}(u) + \kappa^{-1} \bar{D}(u)$ is block diagonal. In the periodic case $\kappa = 1$ its $2 \times 2$ block at $M = 1$ becomes diagonal by $\mathfrak{sl}_2$ symmetry, as the irreps $V_3$ and $V_1$ each occur once in $\mathcal{H}$.

For generic $\theta_1, \theta_2$ the representation of the Yangian is irreducible, unlike for $\mathfrak{sl}_2$. To see this explicitly, notice that any subspace invariant under Yangian action has to lie inside either $\mathfrak{sl}_2$ irrep $V_3$ or $V_1$ of $\mathcal{H}$. From the above we read off

$$\bar{B}(u)|1,1\rangle = \sqrt{2}\,i(u-u_0)|1,0\rangle - \frac{i}{\sqrt{2}}(\theta_1 - \theta_2 - i)|0,0\rangle,$$
$$\bar{B}(u)|0,0\rangle = \frac{i}{\sqrt{2}}(\theta_1 - \theta_2 + i)|1,-1\rangle. \tag{D.5}$$

For generic $\theta_i$ the $B$-operator mixes $V_3, V_1$, so there are no invariant subspaces for the Yangian.

Note from (D.5) that $|0,0\rangle$ becomes an eigenvector of the $B$-operator iff

$$\theta_2 = \theta_1 + i. \tag{D.6}$$

In this case $|0,0\rangle$ is also an eigenvector of $\bar{A}$, $\bar{C}$ and $\bar{D}$, so $V_1$ becomes Yangian invariant (fusion into singlet). Yet $V_3$ still is not invariant, cf. (D.5). Thus the Yangian representation on $\mathcal{H}$ is reducible but indecomposable. Also note from (D.5) that, at the special Bethe root $u_0 = (\theta_1 + \theta_2)/2$ from (57), the $B$-operator sends the vacuum $|0\rangle = |1,1\rangle = |\uparrow\uparrow\rangle$ to a multiple of $|0'\rangle = |0,0\rangle \in V_1$. This illustrates several parts of the discussion in Sections 2.4.1 and 2.4.2.

Similarly, by (D.5) we have $\bar{B}(u)|1,1\rangle \in V_3$ iff

$$\theta_2 = \theta_1 - i, \tag{D.7}$$

and one can check that $V_3$ becomes an invariant subspace for the Yangian (fusion into triplet). This time $V_1$ is not invariant, see (D.5), and we again have a reducible but indecomposable Yangian representation. Observe that this time $\bar{B}(u_0)|0\rangle = 0$ at the special fixed root.

### D.2   Fusion into singlet for $L = 4$

Since we are most interested in the case of fusion into a singlet let us illustrate the discussion from Section 2.4.2 with another example. To see the features related to the Bethe Ansatz we take $L = 4$, with Hilbert space

$$\mathcal{H} = V_2^{\otimes 4} \cong V_5 \oplus 3\,V_3 \oplus 2\,V_1, \qquad \text{for} \quad \mathfrak{sl}_2, \tag{D.8}$$

where the quintet contains the reference state $|0\rangle$, and there are three triplets and two singlets.

Let us fuse the two middle sites, by taking inhomogeneities

$$\theta_3 = \theta_2 + i, \qquad \text{with } \theta_1, \theta_2, \theta_4 \text{ in general position.} \tag{D.9}$$

The invariant subspace is

$$V_{\text{inv}} = \Pi_{23}^-(\mathcal{H}) \cong V_2 \otimes V_1 \otimes V_2 \cong V_3 \oplus V_1, \qquad \text{for} \quad \mathfrak{sl}_2. \tag{D.10}$$

Let us denote the copies of the triplet and singlet inside $V_{\text{inv}} \subset \mathcal{H}$ by $V_{\text{inv},3}$ and $V_{\text{inv},1}$. The complement $\Pi_{23}^+(\mathcal{H}) \cong V_2 \otimes V_3 \otimes V_2$ is not invariant: the $B$-operator sends $|0\rangle \in \Pi_{23}^+(\mathcal{H})$ to (C.1), which spans the $M = 1$ sector and thus has nontrivial overlap with $V_{\text{inv},3}$. As a Yangian representation $\mathcal{H}$ is therefore reducible but indecomposable, as illustrated in Figure 3.[20]

We will describe the construction of the eigenstates of the transfer matrix by algebraic Bethe Ansatz $\bar{B}(u_1)\cdots\bar{B}(u_M)|0\rangle$.[21] For simplicity we consider the periodic case $\kappa = 1$, for which the decomposition (D.8) describes the degeneracies of the transfer matrix eigenvalues, so there are six distinct eigenvalues, corresponding to six eigenvectors with highest weight for $\mathfrak{sl}_2$, occuring at the sectors with $M \leqslant 2$ spins $\downarrow$. In the Bethe equations one of the factors in the numerator and denominator on the left-hand side cancel, yielding

$$\frac{u_m - \theta_1 + i/2}{u_m - \theta_1 - i/2} \frac{u_m - \theta_4 + i/2}{u_m - \theta_4 - i/2} \frac{u_m - \theta_2 + i/2}{u_m - \theta_2 - 3i/2} = \prod_{n(\neq m)}^{M} \frac{u_m - u_n + i}{u_m - u_n - i}. \tag{D.11}$$

For $M = 1$ the algebraic Bethe Ansatz reads (C.1). The right-hand side of (D.11) is unity and we obtain a degree-two polynomial equation for the Bethe root, with solutions that we denote by $u_{1,\pm}$. Thus we obtain two states

$$\bar{B}(u_{1,\pm})|0\rangle, \tag{D.12}$$

---

[20]In particular, if we plug into $B$ the fixed Bethe root it gives a state lying entirely in $V_{\text{inv},3}$, see (D.13).

[21]The following is based on numerics, but we expect our findings to hold for generic $\theta_1, \theta_2, \theta_4$.

that are the highest-weight vectors in the triplets contained in $\Pi_{23}^+(\mathcal{H})$. The remaining $\mathfrak{sl}_2$ highest-weight state with $M = 1$ is

$$|0'\rangle = \bar{B}(u_0)|0\rangle \in V_{\text{inv},3}, \qquad u_0 = \frac{\theta_2 + \theta_3}{2} = \theta_2 + \frac{\text{i}}{2}, \tag{D.13}$$

as expected from the discussion in Section 2.4.3. It obeys the Yangian highest-weight conditions $\bar{C}(u)|0'\rangle = 0$ and is an eigenvector of both diagonal elements of the monodromy matrix:

$$\bar{A}(u)|0'\rangle = \frac{u - u_0 - \text{i}}{u - u_0}\bar{Q}_{\boldsymbol{\theta}}^+(u)|0'\rangle, \qquad \bar{D}(u)|0'\rangle = \frac{u - u_0 + \text{i}}{u - u_0}\bar{Q}_{\boldsymbol{\theta}}^-(u)|0'\rangle. \tag{D.14}$$

It remains to discuss the two singlets from (D.8) at $M = 2$. One is of the standard form

$$\bar{B}(u_1)\bar{B}(u_2)|0\rangle, \tag{D.15}$$

where $u_1, u_2$ solve the Bethe equations (D.11) with $M = 2$. Notice there is only one admissible solution of those Bethe equations with $M = 2$, i.e. without repeated or infinite Bethe roots. The last singlet state in (D.8) spans $V_{\text{inv},1} \subset V_{\text{inv}}$. As expected, we can obtain it using the $B$-operator acting on (D.13) with the Bethe root determined by the reduced Bethe equations (60), which read

$$\frac{u_1' - \theta_1 + \text{i}/2}{u_1' - \theta_1 - \text{i}/2}\frac{u_1' - \theta_4 + \text{i}/2}{u_1' - \theta_4 - \text{i}/2} = 1. \tag{D.16}$$

Thus we obtain this last singlet state as

$$\bar{B}(u_1')|0'\rangle = \bar{B}(u_0)\bar{B}(u_1')|0\rangle, \qquad u_1' = \frac{\theta_1 + \theta_4}{2}. \tag{D.17}$$

We see that all $\mathfrak{sl}_2$ highest-weight states are given by the algebraic Bethe Ansatz. (Here $u_1'$ happens to have the same form as $u_0$, but this is a coincidence for low $L$, unrelated to any cancellations like in Appendix B or any other special features in the presence of fusion.)

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
