# Peer review of "Bethe ansatz inside Calogero-Sutherland models"

_SciPost Physics, doi:SciPost Phys. 18, 035 (2025)_

## Round 2 · Referee Report · Anonymous (Referee 1) · 2024-11-8

Strengths

  1. The paper establishes a new way to apply the Algebraic Bethe Ansatz for the models with long-range interactions
  2. The authors propose a new approach permitting to use the inhomogeneous Heisenberg chain to treat spin Calogero-Sutherland and Haldane-Shastry model.
  3. Clarity of the presentation and numerous examples making the paper easily readable.

Weaknesses

None

Report

The paper treats the relations of the inhomogeneous Heisenberg spin chain with long-range interacting integrable models such as the spin Calogero-Sutherland model and the Haldane-Shastry model. It has long been known that adding inhomogeneity parameters makes the spin chain Hamiltonians non-local, but it is an important breakthrough to use these models to describe long-range interacting systems. I would also like to emphasise the very high clarity of the presentation and the numerous examples used to illustrate the main points.

Requested changes

There are (very few) typos and some points to clarify: 1. On page 8 the phrase "one should take all the solutions etc" should be probably replaced by "one should take all the admissible solutions ..." with proper explanation what is admissible solution. 2. On page 11 the last statement of the subsection 2.3.3 should be better explained: why the degree of $\tilde{Q}$ is different without twist and what are the roots of this second solution. It seems to be a quite important piece of information for some of the subsequent results. 3. Page 19 (beginning of Section 3) mode -> model 4. Page 22 On the fermionic space space -> On the fermionic space

Recommendation

Publish (easily meets expectations and criteria for this Journal; among top 50%)

---

## Round 2 · Referee Report · Anonymous (Referee 2) · 2024-11-17

Strengths

1- It is innovative and solves a relevant and technically difficult problem in the field. 2- It takes a well known procedure in the literature and use it in a completely innovative way to answer a challenging question: the diagonalisation of long-range integrable models. 3- It is extremely well written. Sections clearly built on previous ones. 4-The introduction on inhomogeneous spin chains is interesting by itself.

Weaknesses

None

Report

Integrable long-range spin chains have received considerable attention in the last two decades due to their applications in four-dimensional gauge theories. From an algebraic point of view, however, the open questions are numerous.

The authors of the  manuscript "Bethe Ansatz inside Calogero-Sutherland models" successfully solved a very important of these open questions, bringing in addition, new relevant insights to the discussion.  More specifically they used the Bethe ansatz method, in a version inherited from the inhomogeneous Heisenberg spin chain (which is by itself a long-range spin chain), to study both the trigonometric quantum spin-Calogero-Sutherland model and the Haldane-Shastry spin chain.

The Introduction gives a clear and accurate description of the current state of the literature in the field. The authors appropriate refer to existent literature and explain how the current manuscript fits into the picture.

Section 2 contains an introduction to the Heisenberg spin chain and includes elements rarely discussed together in other sources. It is very well written and together with all the appendices, can easily become a preferred source of information for anyone wanting to learn the topic. In particular, the whole section about fusion (sec. 2.4) is innovative in its presentation. Additionally, I had never seen the proof that "...any inhomogeneities can be exchanged by a similarity transformation..." presented so clearly. A second family of conserved charges is also presented.

Section 4 contains the main new results of the paper. The authors find a new Bethe ansatz eigenbasis for both the spin-Calogero-Sutherland model and the Haldane-Shastry chain. The results for the latter are obtained via freezing.  This Bethe ansatz diagonalizes, in particular, the new conserved charges of these models constructed in this manuscript. This section is written building on sections 2 and 3. This is particularly useful, since those sections are very clear. With this, the authors easily guide the reader through their construction.

"Bethe Ansatz inside Calogero-Sutherland models" is very well written. It is original and it solves the very relevant question of constructing the Bethe ansatz for two long-range spin models, constructing in the process new charges for these models.  In particular, I would like to highlight that the authors did this by taking  the addition of inhomogeneities in the chain, which is a very well known procedure  in the literature, and using it in a new and very creative way.  
Therefore, the manuscript easily meets the expectations and criteria for SciPost Physics and I recommend it to publication.

Requested changes

I found a few typos and have some minor suggestions. Please see comments/suggestions in attached file.

Attachment

Recommendation

Publish (easily meets expectations and criteria for this Journal; among top 50%)

---

## Round 2 · Referee Report · Anonymous (Referee 3) · 2024-11-28

Strengths

1- Interesting new results on the algebraic Bethe ansatz analysis to the spin Calogero-Sutherland model. 2- The new approach has potential for various generalizations. 3- The manuscript is clearly written

Weaknesses

None that I can see.

Report

In this paper, the authors study the spin Calogero-Sutherland model. While this model is well-known to be integrable, the underlying integrability does not always fit well with the usual tools which have proven their worth in short range models, such as the quantum Heisenberg spin chain. The main technical achievement in this paper is the use of inhomogeneous generalizations of the Heisenberg chain --which can be treated using Algebraic Bethe ansatz-- to understand the long range model. By freezing, the results are also relevant to the study of the Haldane-Shastry model. There are also several directions for generalization as well as applications of their method.

Overall this is an interesting paper, and it is very well written. All arguments and calculations are carefully spelled out. The flow of the paper is also very natural and easy to follow. All inter-relations between the models, as well as the various known limits are carefully discussed. The relevant literature is appropriately quoted.

I have little to add, in my opinion the paper easily meet the expectations for a good paper in Scipost Physics, I recommend publication as it is.

I found two typos:
-in section 3, page 19, replace "integrable mode" by "integrable model".
-at the bottom of Page 24, replace "mondromy" by "monodromy".

Requested changes

None

Recommendation

Publish (easily meets expectations and criteria for this Journal; among top 50%)

---

## Editorial Decision

published